# Reveal the relation between spatial patterns of rainfall return levels and landslide density

Slim Mtibaa[1], Haruka Tsunetaka[1]

[1]Department of Disaster Prevention, Meteorology and Hydrology, Forestry and Forest Products Research Institute, Tsukuba city, 305-8687, Japan

*Correspondence to*: Slim Mtibaa (mtibaaslim@ffpri.affrc.go.jp)

**Abstract.** It is known that the spatial rainfall pattern can mark landslide distribution across the landscape during extreme triggering events. However, the current knowledge of rainfall controls on this distribution remains limited. Here, to reveal what rainfall characteristics control landslide spatial distribution, we explore the spatiotemporal pattern of a rainfall event that triggered over 7,500 landslides (area $\approx 10^0$–$10^4$ m$^2$) at a regional scale with an area of $\approx 400$ km$^2$ in Japan. Using a 5-km resolution radar-driven and gauge-adjusted hourly precipitation dataset with 32 years of records, we compared rainfall return levels for various time ranges from 1 to 72 h and landslide density in each grid cell of the precipitation dataset ($\approx 25$ km$^2$). The results show that, even if local slope distributions within the grid cells are comparable, the number of landslides in a $\approx 25$ km$^2$ grid cell was substantially high when rainfall return levels exceeded the 100-year return period in all examined timespans (i.e., 1–72 h). In contrast, when only specific-duration rainfall intensities (e.g., 6–48 h) exceeded the 100-year return level, the landslide density in corresponding grid cells tended to be low. Consequently, the landslide density increased with the increase in rainfall return levels of various timespans rather than a specific rainfall intensity, such as downpours for a few hours or long-term cumulative rainfall for several days. Moreover, with the increase in the landslide density, the number of relatively large landslides exceeding $\approx 400$ m$^2$ increased. Therefore, the spatial differences in rainfall return levels potentially constrain the density of total landsliding and relatively large landslides. In this sense, whether rainfall intensities reach high return levels rarely experienced in a wide timespan ranging from a few hours to several days is one of the key determinants of the spatial distribution of landslides and the extent of related hazards.

## 1 Introduction

Landslides are natural geomorphic processes driving long-term landscape evolution (Korup et al., 2010), which may impose substantial changes in hillslope and fluvial systems and significant human and economic losses (Froude and Petley, 2018; Jones et al., 2021). Rainfall is the most common trigger of landslides (Sidle and Bogaard, 2016). Although rainfall may provoke individual landslides with localized impacts, large-scale extreme rainfall events often induce numerous landslides widely spread over the landscape (Emberson et al., 2022). In such cases, landslide impacts span the spatial extent

of the triggering event, and their significance depends on the location and magnitude (i.e., number and size) of triggered

landslides (Medwedeff et al., 2020; Milledge et al., 2014; Benda and Dunne, 1997). Therefore, revealing rainfall controls on landslide spatial distribution through investigating the relationship between rainfall and landsliding is fundamental for assessing landscape changes and supporting hazard prediction efforts.

A well-established method for linking landslide occurrence to rainfall or hydrological characteristics (e.g., intensity, duration, soil moisture) is the use of rainfall thresholds (Guzzetti et al., 2008; Caine, 1980; Saito et al., 2010) and recently

hydro-meteorological thresholds (Bogaard and Greco, 2018). These empirical thresholds offer a straightforward way to predict whether landslides will occur in the future. However, they cannot quantify the magnitude of landslides. Therefore, multiple studies attempted to constrain quantitative spatial relationships between landslide distribution, often described as density (e.g., number/km$^2$ or area/km$^2$), and dynamic explanatory variables that provide proxies for the critical rainfall conditions triggering landslides. Typically, these studies aimed at identifying the key rainfall variable(s) that drive

landsliding by relying upon regression analysis and specific landslide records (i.e., a catalog of individual landslide information (e.g., Gao et al., 2018), detailed landslide inventories triggered by single or multiple rainfall events (e.g., Marc et al., 2018; Chang et al., 2008)).

So far, we still lack information on the best rainfall variable(s) constraining the landslide spatial pattern during rainfall events. Some works showed increased landslide density with the increase in total rainfall amount, rainfall duration, the

maximum rainfall amount for short durations (e.g., 3, 12, 24 h), or antecedent rainfall (Marc et al., 2018; Chen et al., 2013; Chang et al., 2008; Dai and Lee, 2001; Abanco et al., 2021). Other studies demonstrated that normalized rainfall amounts for specific timespans (e.g., 2, 24, 48 h) by the mean annual precipitation (Ko and Lo, 2016) or the 10-year return period rainfall amount (Marc et al., 2019), which explain the landscape coevolution with local climate (Benda and Dunne, 1997; Iida, 1999), are better predictors for landsliding.

On the other hand, these statistical relationships allow the development of rainfall-based empirical models for predicting the number of landslides likely to be triggered by future rainfall events (e.g., Chang et al., 2008). However, their development and extrapolation to other regions are challenging. Constraining any spatial relationship requires comprehensive landslide inventories that contain sufficient landslides for an adequate statistical analysis. However, this need is extremely difficult to fulfill (Marc et al., 2018; Emberson et al., 2022). Furthermore, the constrained quantitative

relationships are very sensitive to the landslide records and the characteristics of respective triggering rainfall events used in the statistical analysis. Therefore, they are case-specific and cannot always be extrapolated to predict the number of landslides likely to be triggered by future rainfall events, even in the same region (e.g., Gao et al., 2018).

For a given rainfall event, the return period of any rainfall episode with specific duration and intensity can be assessed using the Intensity-Duration-Frequency (IDF) curves, which are equipotential lines of probabilities linking rainfall

durations and maximum intensities from long-term records (Chow et al., 1988). This information can potentially evaluate whether a rainfall event is likely to cause landslides as a high rainfall return level (i.e., rare rainfall event) is generally

considered a proxy for the critical rainfall conditions triggering landslides (Frattini et al., 2009; Griffiths et al., 2009; Segoni et al., 2015, 2014; Iida, 2004). Several studies showed the usefulness of considering rainfall return levels to indirectly evaluate the potential of a forecast rainfall to trigger landslides without the need for historical landslide records in the targeted region (e.g., Kim et al., 2021; Tsunetaka, 2021; Vaz et al., 2018). Still, the potential relation between the spatial patterns of rainfall return levels and landsliding remains unrevealed.

Clearly, rainfall controls on landslide spatial distribution differ depending on rainfall characteristics and local terrain settings (e.g., Bogaard and Greco, 2018). Even during the same triggering rainfall event, multiple inventories showed discrepancies in landslide occurrence timing and geometric features (e.g., area, volume, and depth) at the catchment (Yamada et al., 2012; Yano et al., 2019; Guzzetti et al., 2004) and hillslope scales (Büschelberger et al., 2022). This suggests that landslides are triggered by disparate rainfall timespans due to different hydromechanical responses of hillslopes to forcing rainfall. If so, then it is reasonable to hypothesize that landsliding can be constrained by the return levels of multiple rainfall timespans. This study focused on an extreme rainfall event that triggered over 7,500 landslides in an area of around 400 km$^2$ in the northern part of the Kyushu region in southern Japan to investigate whether spatial patterns of rainfall return levels govern landslide density. Using a gridded rainfall dataset with a $\approx$ 5-km resolution, we compared rainfall return levels for various time ranges from 1 to 72 h and landslide density in each $\approx$ 25-km$^2$ grid cell to investigate whether the landslide density increase in grid cells where rainfall intensities reach high return levels that are rarely experienced. The present research is expected to provide insights into what rainfall characteristics control landslide spatial distribution and when rainfall may cause high landslide density. Thus, it can have promising implications for supporting hazard prediction efforts and understanding landscape evolution.

## 2 Material and Methods

### 2.1 Study site and landslide characteristics

The study focuses on an area of around 400 km$^2$ in the northern part of the Kyushu region in southern Japan (Fig. 1a). The examined area experienced an extreme rainfall event on July 5 and 6, 2017, caused by a linear mesoscale convective system (Hirockawa et al., 2020), that triggered over 7,500 landslides (Fig. 1a).

If the landslides occurred in a homogeneous regolith, which reduces the likelihood of their link to complex geotechnical site characteristics (Marc et al., 2019), the interpretation of the potential rainfall controls on landslide occurrence would be possible. Indeed, most landslides triggered by the examined rainfall event were shallow, affected mainly the soil mantle, and occurred on forested hillslopes with similar lithological settings (granodiorite and pelitic schist) (Chigira et al., 2018). Accordingly, previous investigations of the importance of multiple predisposing factors (e.g., rainfall, slope, elevation, land cover, etc.) in the occurrence of these landslides using machine learning methods showed the outweighing of rainfall conditions on the other predisposing factors (Dou et al., 2020; Ozturk et al., 2021). Thus, the examined area provides an

adequate test field to investigate the rainfall controls on landslide density because at least the land cover and lithological settings of hillslopes can be deemed relatively homogenous.

Our research relied on the landslide inventory prepared by the Ministry of Land, Infrastructure, Transport, and Tourism of Japan from orthophotos of 0.1-m resolution and digital elevation models (DEM) of 1-m resolution acquired by Airborne Laser Scanning in July 2017 (i.e., immediately after the landslide occurrence). The mapping method of landslide scars involves three steps. The first step identifies bare land hillslopes as landslides and delineates them manually from the orthophotos. The second step rectifies the delineated landslide scars using DEM data acquired after the disaster and maps

them as polygons. The third step compares these polygons to satellite and aerial images dated before July 2017 to exclude landslides that formerly occurred in the region. The inventory counts 7,676 polygons identifying widespread landslides in the examined area (Fig. 1b). These polygons represent only landslide source areas (scars) and omit runout zones.

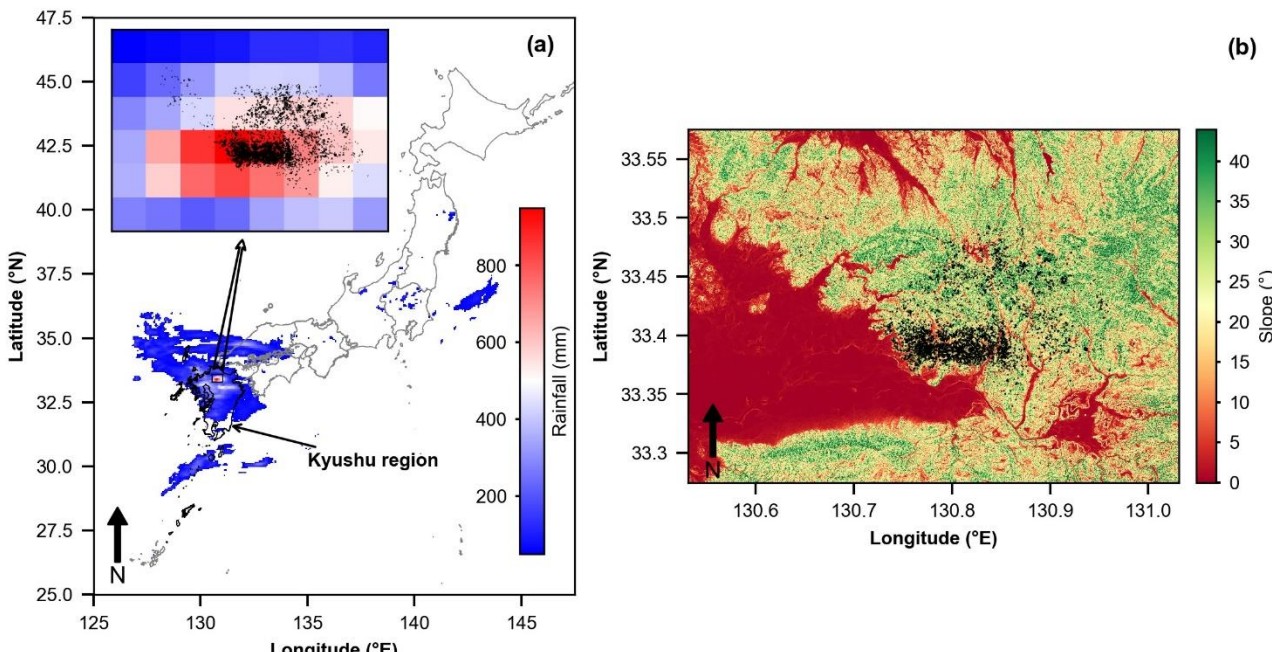

**Figure 1: (a) Cumulative rainfall for 5 and 6 July 2017 (> 50 mm) and location of triggered landslides (black polygons in the**
**inset figure). (b) Distribution of the landslides (black polygons) over the Slope map of the affected region.**

We investigated landslide size characteristics by examining the frequency-area distribution (FAD), which plots landslide sizes (i.e., measures of the area) with corresponding frequencies (Malamud et al., 2004). The FAD can determine whether the landslide inventory follows the fundamental property of landslides (Hovius et al., 1997). For the landslide inventory this study relied on, the FAD exhibited a rollover (i.e., the peak point of the distribution) at around $10^2$ m$^2$, below which

the frequency of small landslides decreases, and a cutoff point of 439 m$^2$ (Fig. 2), which was derived using the method of Clauset et al., (2009). The frequency distribution of landslides with area size exceeding the cutoff (area > 439 m$^2$), which

accounted for 28.12 % of the total inventory and referred to, hereafter, as medium and large landslides, fitted a power-law function with the scaling parameter ($\beta$) of 2.26. This exponent is within the typical range of 2–3 derived by other landslide inventories (e.g., Guzzetti et al., 2002; Marc et al., 2018) and suggests that the small landslides were more frequent than medium and large landslides (area > cutoff point of 439 m$^2$) during the studied event. Accordingly, it is important to note that the landslide inventory follows the fundamental properties of landslides, as the FAD can fit an inverse gamma distribution with a right tail that decays as a power law (Stark and Hovius, 2001). Considering the high resolution of DEM and orthophotos used for constructing the examined landslide inventory, which is significantly lower than the cutoff point and allowed capturing the geometric features of landslides with size in the order of 0.02 m$^2$, it is evident that the observed divergence was due to physical processes rather than under-sampling of small landslides (Frattini and Crosta, 2013; Medwedeff et al., 2020).

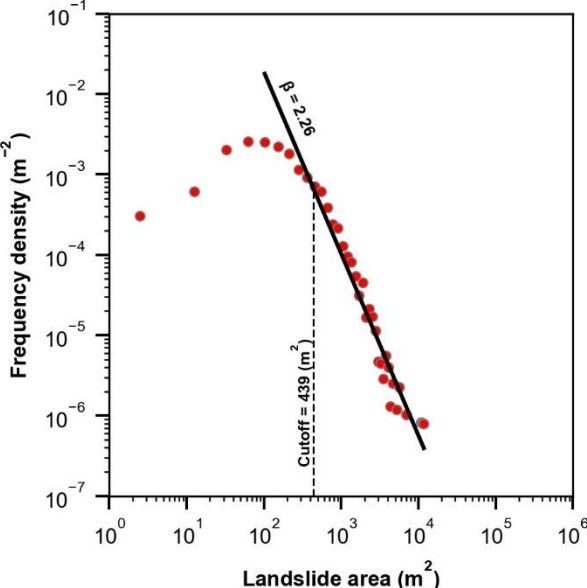

**Figure 2: Non-cumulative frequency area distribution of the landslide inventory.**

Additionally, we quantified landslide angles as the median slope at landslide scars derived from the analysis of a 10-m resolution DEM, which was developed by the Geospatial Information Authority of Japan (GSI) from 1:25,000-scale topographic maps dated before the disaster (Fig. 1b). For landslides with an area smaller than 100 m$^2$ (i.e., DEM pixel size), the slope value of the pixel was taken as landslide angle. The landslide angles ranged between 0.45° and 51.03° (median = 27.20°). More than 90 % of the triggered landslides were associated with hillslopes of more than 16.26° slope (Supporting information, Fig. S1).

## 2.2 Rainfall data and processing methods

### 2.2.1 Rainfall data

We employed the radar/rain gauge analyzed (R/A) precipitation dataset to examine the spatiotemporal pattern of the triggering rainfall and derive the return levels of rainfall intensities for multiple timespans in the Intensity Duration Frequency (IDF) curves. The R/A dataset is a gridded hourly precipitation product developed by the Japan Meteorological Agency (JMA) based on 5-minutely reflected echo intensities and doppler velocities of 46 C-band radars (Nagata, 2011). The processing algorithm of this product includes three steps. First, accumulated radar echo intensity data were processed by a quality control algorithm for correcting precipitation observation errors attributed to various meteorological, topographic, and technical factors (e.g., beam blockage, ground clutter, anomalous beam propagation, and range effects) (Makihara, 2000). Subsequently, the hourly accumulated corrected radar data were adjusted to rainfall measurements obtained from local rain gauges to produce accurate Quantitative Precipitation Estimates (QPE). Finally, the calibrated QPE from the 46 radars were processed and assembled to derive nationwide hourly precipitation maps that compose the R/A product (Makihara, 2000; Nagata, 2011). This correction and processing scheme made the R/A dataset the most reliable long-term precipitation data over the Japanese archipelago. Accordingly, it has often been used as referential data for analyzing localized heavy rainfall (e.g., Kato, 2020; Hirockawa et al., 2020; Saito and Matsuyama, 2015), evaluating precipitation forecasts and estimates (e.g., Kubota et al., 2009; Iida et al., 2006; Yin et al., 2022), and constraining empirical relationships between rainfall information and landslide occurrence (e.g., Saito et al., 2010; Marc et al., 2019; Ozturk et al., 2021).

In this study, we relied on the R/A data for 1988–2019. The product provides hourly adjusted rainfall estimates with a spatial resolution of ≈ 5 km (1988–2001), ≈ 2.5 km (2002–2005), and ≈ 1 km (from 2006) (Mtibaa and Asano, 2022). Therefore, for homogeneity reasons, we downscaled the data from 2002 to ≈ 5 km spatial resolution (same as the resolution of the 1988–2001 dataset) using the method recommended by Nagata and Tsujimura (2006). We selected this method because it produced homogenous maximum hourly and daily rainfall time series based on the homogeneity tests applied by Urita et al. (2011) and Saito and Matsuyama (2015). It spatially averages the 1 km product to 2.5 km spatial resolution and downscales the 2.5 km product to 5 km spatial resolution by selecting the maximum value of the four 2.5 km grid cells. Although the downscaling stage degrades the spatial details of rainfall events, it is unavoidable in this study due to the requirement of long-term rainfall data in investigating rainfall return levels. Still, the downscaled R/A dataset (i.e., 5-km resolution) can capture spatial rainfall patterns over the examined region as it could sufficiently resolve mesoscale convective systems that resulted in most heavy rainfall events in Japan (Hirockawa et al., 2020).

### 2.2.2 Rainfall processing methods

As stressed in the Introduction, owing to the hillslope-scale variation in the effective rainfall needed for triggering landslides, using multiple rainfall durations is crucial for elucidating the relation between the potential of these rainfall timespans to trigger landslides and the spatial pattern of landslide density. Because the correct timing of respective landslide occurrence is unknown and probably different within each grid cell of the R/A precipitation dataset, setting a standardized rainfall period covering a combination of disparate rainfall timespans from short to long duration deemed responsible for

triggering landslides is required for comparisons between spatial distributions of rainfall and landslide density. In this study, the 72 h that accumulated the maximum rainfall during the examined rainfall event was used as the standardized rainfall period ($P_{std}$), as suggested by Tsunetaka. (2021). We assumed that the various landslides experienced in our study area occurred within this period. This assumption was based on the fact that the studied event brought unprecedented rainfall amount that outweighs the possible effects of antecedent rainfall on landslide occurrence (Marc et al., 2019; Guthrie

and Evans, 2004). The temporal rainfall pattern was subsequently examined by computing the maximum rainfall intensity (rainfall intensity maxima) for multiple timespans (1, 2, 3, 6, 12, 24, 48, and 72 h) within the $P_{std}$ for all R/A grid cells.

To investigate the return levels (i.e., recurrence levels) of these rainfall intensity maxima, we developed the IDF curves that statistically fit the annual maxima series (AMS) of rainfall intensities observed over 1–72 h. We extracted the rainfall AMS from the 32-year (from 1988 to 2019) R/A precipitation dataset. Then, we used the Gumbel distribution based on the

L-moments method (Hosking, 1990) to fit the extracted rainfall AMS due to its few shape parameters that may reduce the estimation uncertainty (Frattini et al., 2009). Such a statistical model assumes an asymptotic behavior of the rainfall dataset and a stationarity in the rainfall AMS. To assess the ability of the estimated distributions to represent the extracted rainfall AMS, we used the Kolmogorov-Smirnov (KS) test, which examines the goodness of fit between the estimated and observed cumulative distributions. Here, the null hypothesis assumes identical distributions. Therefore, the *p-value* calculated using

an asymptotic distribution of the KS test statistic should be less than a significance level of 5 % to reject the null hypothesis. Although the Gumbel distributions may well fit the observed rainfall AMS based on the KS test, this does not mean that the derived IDF curves do not shift over time (i.e., stationary) due to climate change (Slater et al., 2021). It is, therefore, crucial to test the stationarity assumption in the Gumbel model parameters by assessing the existence of trends in rainfall AMS during the examined period. To this end, we employed the Mann-Kendall and Sen's slope tests, two non-parametric

statics frequently applied in hydro-meteorology for trend analysis (e.g., Yan et al., 2018). The Mann-Kendall test assesses the significance of trends in rainfall (Mann, 1945; Kendall, 1975), while Sen's slope test quantifies the magnitude of these trends if exist (Sen, 1968). The null hypothesis of the Mann-Kendall test assumes no trends. Therefore, a *p-value* less than a significance level of 5 % would imply the existence of a significant trend in rainfall AMS.

### 2.3 Investigating rainfall controls on landslide spatial distribution

### 2.3.1. Landslide density

The spatial distribution of triggered landslides over the study area can be described as a spatial variation of landslide density (i.e., number/km$^2$). Landslide density is generally calculated by counting the number of landslides that occurred within a specific area. Here, because we intended to reveal the potential control of rainfall return levels for multiple timespans derived from the R/A dataset on the variation of landslide density, we used the R/A grid cell ($\approx 25$ km$^2$) as a sliding window to calculate landslide density. To count the number of landslides that occurred within each R/A grid cell, we converted the polygons data of landslide scars to points locating the centroid of each polygon. These numbers are generally biased by the non-uniformly distributed topographic features (i.e., hills, mountains, plains, lakes) within the different R/A grid cells because landslides commonly occur in hilly and mountainous areas rather than plains (Lombardo et al., 2021). To avoid such a possible bias, landslide density was calculated as the number of landslides within each R/A grid cell divided by the area of the R/A grid cell where the slope is higher than a threshold angle ($S_{threshold}$) assumed to be a minimum angle to allow landsliding. $S_{threshold}$ defines the threshold angle above which 90 % of landslides occurred (Prancevic et al., 2020) and was determined as 16.26° based on the DEM data analysis (Fig. S1).

Although medium and large landslides (landslides with area size exceeding the cutoff point of the FAD (439 m$^2$)) counted only 28.12 % of the total landslides, their areas represented more than 70 % of the total landsliding area (i.e., the total scar areas of the triggered landslides). Therefore, it is interesting to investigate rainfall controls on the density of total and only medium and large landslides. Accordingly, we computed two landslide density metrics, total landslide density (TD) and only medium and large landslide density (MLD), as the number of landslides per unit area (km$^2$), for each R/A grid cell using the following equations (1) and (2). Note these metrics represent averaged landslide density within the R/A grid cells.

$$TD = \frac{Total\ number\ of\ all\ landslides\ within\ an\ R/A\ grid\ cell}{A_{threshold}} \qquad (1)$$

$$MLD = \frac{Number\ of\ medium\ and\ large\ landslides\ within\ an\ R/A\ grid\ cell}{A_{threshold}} \qquad (2)$$

Where, $A_{threshold}$ is the area in km$^2$ of an R/A grid cell where the slope $> S_{threshold}$ (i.e., 16.26°).

### 2.3.2. Relationships between the spatial pattern of landslide density and rainfall information

Similar to previous studies (e.g., Chang et al., 2008), our investigation started by evaluating the statistical correlations between calculated landslide density metrics (TD and MLD) and rainfall intensity maxima for multiple timespans (1–72 h). We used Spearman's rank coefficient ($\rho$) to measure the non-parametric monotonicity of these relationships. In doing so, we intended to explore whether the developed statistical relationships can explicitly explain the rainfall controls on landslide density. Subsequently, we compared the variation in rainfall intensity maxima and their return levels and landslide density at the R/A grid cell scale.

Although the use of $A_{threshold}$ as a normalization method for calculating TD and MLD suppresses the influence of the non-uniformly distributed topographic features within the different R/A grid cells, still, these metrics can be biased by the non-uniformly distribution of local slopes within the $A_{threshold}$ as landslide occurrence also depends on hillslope steepness (Prancevic et al., 2020). Therefore, it is crucial to focus on R/A grid cells with comparable local slope distributions to explicitly investigate the potential control of rainfall intensity maxima and their return levels on landslide density. To this end, we first tested the differences in local slope angle distribution within $A_{threshold}$ of the different R/A grid cells using the Kruskal-Wallis test (Kruskal and Wallis, 1952). Then, we employed Dunn's nonparametric pairwise test (Dunn, 1961) with a Bonferroni correction for the *p-value* for detecting the R/A grid cells with similar mean rank sums of slopes within $A_{threshold}$ (similar slope conditions). Here, the null hypothesis assumes no significant differences in the distribution of slope angles within the $A_{threshold}$ of the R/A grid cells. Therefore, the *p-value* should be higher than a significant level of 5 % to accept the null hypothesis (Dinno, 2017). Accordingly, the pairwise R/A grid cells, where Dunn's test accepts the null hypothesis, would be ideal examples for comparing the relation between rainfall intensity maxima and their return levels and the variation of landslide density metrics.

## 3 Results

### 3.1 Relationship between landslide density and rainfall intensity maxima

A line-shaped band of high rainfall intensity maxima matched the overall spatial pattern of triggered landslides (Fig. 3), indicating that the spatial distribution of rainfall intensities constrains the landslide distribution. These maxima exhibited substantial differences at the R/A grid cell scale, suggesting spatial disparity in the characteristics of the temporal rainfall pattern. The total triggered landslides were distributed within 23 R/A grid cells with a TD varied between 0.05 and 105.63 landslides/km$^2$ and an MLD ranging between 0.00 and 36.26 landslides/km$^2$ (Fig. 3). More than 65 % of the total landslides occurred within only three R/A grid cells with a TD of 35.61, 103.88, and 105.63 landslides/km$^2$. The MLD values in these R/A grid cells were 11.98, 36.26, and 28.03 landslides/km$^2$, respectively, indicating the highest number of medium and large landslides occurred during the triggering event. From a statistical point of view, Spearman's rank correlation coefficients (Table 1) showed significant monotonic positive relationships between all computed rainfall intensity maxima and TD ($0.62 < \rho < 0.80$) and MLD ($0.68 < \rho < 0.84$) at the 1 % level. However, these relationships did not necessarily mean that landslide density increases with increased rainfall intensity maxima, as we observed R/A grid cells with comparable rainfall intensity maxima but different TD and MLD (e.g., Fig. S2n and r). Therefore, rainfall controls on landslide density cannot be explicitly grasped from the developed statistical relationships.

The 23 R/A grid cells, where the triggered landslides were distributed, exhibited significant non-uniformly distributed local slopes within $A_{threshold}$, as shown in Fig. S3, and confirmed by the rejection of the null hypothesis of the Kruskal-Wallis

test (*p-value* < 0.05). Applying Dunn's post hoc test, we could idealize three pairs of R/A grid cells with comparable slope distributions within $A_{threshold}$, as Dunn's test could not reject the null hypothesis (Table S1). These three pairs of R/A grid cells were referred to as P1, P2, and P3 and focused on hereafter to explicitly investigate the relation between rainfall intensity maxima and landslide density (Fig. 4). Note we excepted three R/A grid cells where most landslides occurred in areas affected by anthropogenic activities (e.g., slopes surrounding cropland and paddy field) from the Dunn's post hoc test.

Despite the similarity in local slope distributions, the differences in landslide density (TD and MLD) between the paired R/A grid cells in P1 and P2 were well distinguishable ($\approx 700$ times and $\approx 70$ times, respectively). In P1, the rainfall intensity maxima observed over the R/A grid cell that experienced high landslide density (TD = 35.61 and MLD = 11.98 landslide/km$^2$) were 1.5 to 1.7 times higher than those observed in the low landslide density R/A grid cell (Fig. 4a). Similarly, the differences in rainfall intensity maxima over the paired R/A grid cells in P2 varied between 1.7 to 3.3 times of rainfall intensity (Fig. 4b). Thus, some paired R/A grid cells with comparable local slope distributions showed that landslide density increased with the increase in rainfall intensity maxima.

Importantly, even with comparable rainfall intensities and slope distributions, landslide density over two R/A grid cells could be different (Fig. 4c). Unlike the observations in P1 and P2, rainfall maxima recorded for 12–72 h over the two R/A grid cells in P3 (Fig. 4c) were similar. The R/A grid cell with higher landslide density experienced little higher rainfall intensity maxima for 1–6 h timespans than those recorded in the R/A grid cell with lower landslide density. But, the differences in these rainfall intensity maxima were slight ($\approx 1.15$ times) compared to those observed between the paired R/A grid cells in P1 and P2. Because P1 and P2 paired two of the R/A grid cells with the lowest landslide density metrics during the examined rainfall event with two of the R/A grid cells with the highest landslide density metrics, the differences in landslide density metrics were much more pronounced than that observed over the R/A grid cells in P3 ($\approx 3.5$ times for TD). However, the R/A grid cell with higher landslide density in P3 indicated the fifth highest TD (20.91 landslides/km$^2$) and MLD (5.65 landslides/km$^2$) in the total of 23 R/A grid cells (Fig. S3), being a sufficiently high landslide density. Given this, the results in P3 indicated that differences in rainfall intensities and slope distributions (i.e., topography) do not necessarily constrain landslide density.

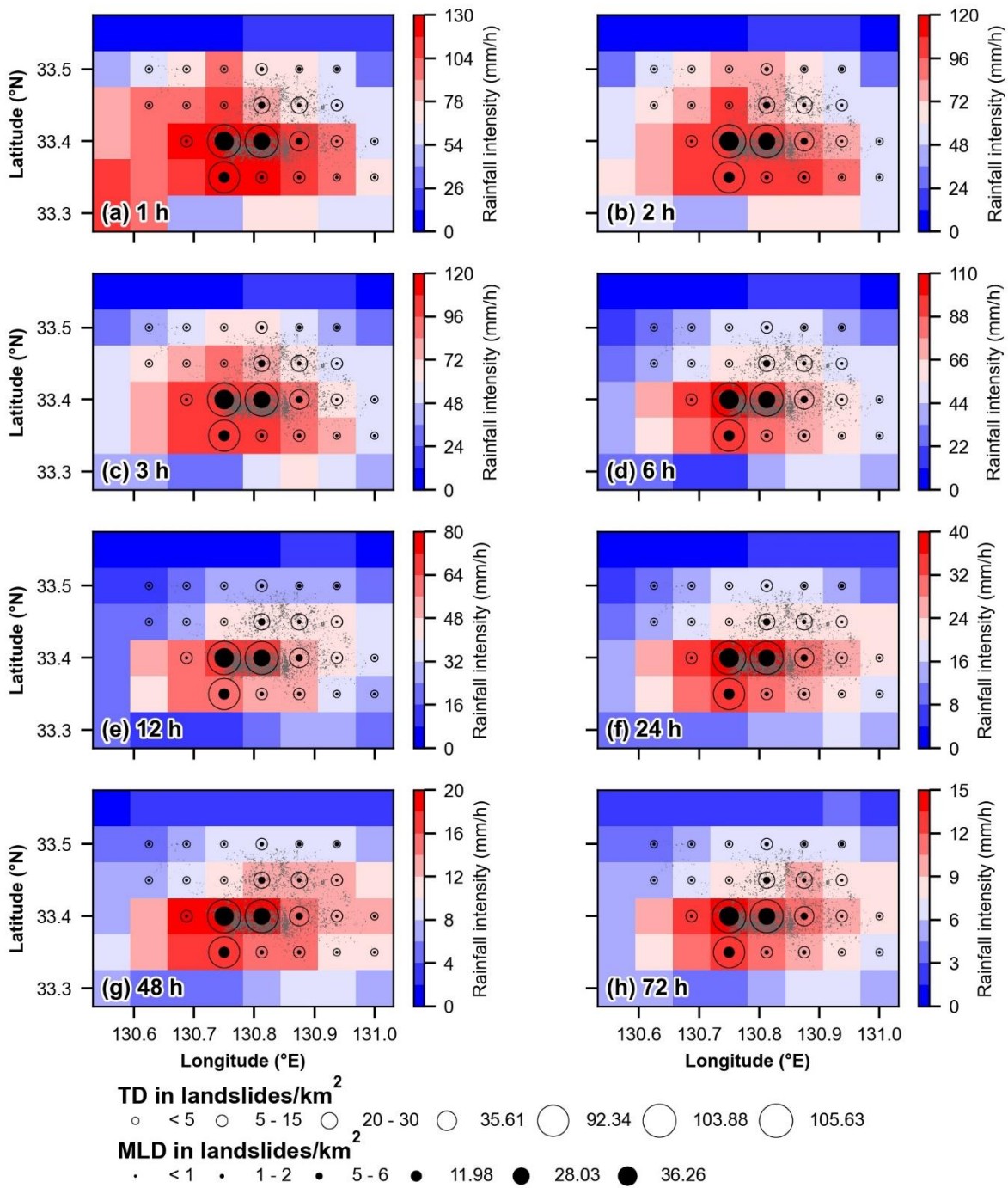

**Figure 3: Spatial distribution maps of rainfall intensity maxima for 1 to 72 h timespans within $P_{std}$ in mm/h, triggered landslides (grey polygons), and landslide density metrics (circles)**

**Table 1: Spearman rank correlation between rainfall intensity maxima and landslide density metrics**

| Rainfall timespan (h) | 1 | 2 | 3 | 6 | 12 | 24 | 48 | 72 |
|---|---|---|---|---|---|---|---|---|
| ρ (TD) | 0.62* | 0.66* | 0.74* | 0.79* | 0.79* | 0.79* | 0.79* | 0.80* |
| ρ (MLD) | 0.68* | 0.71* | 0.77* | 0.84* | 0.82* | 0.81* | 0.81* | 0.82* |

* significant at 1 % level

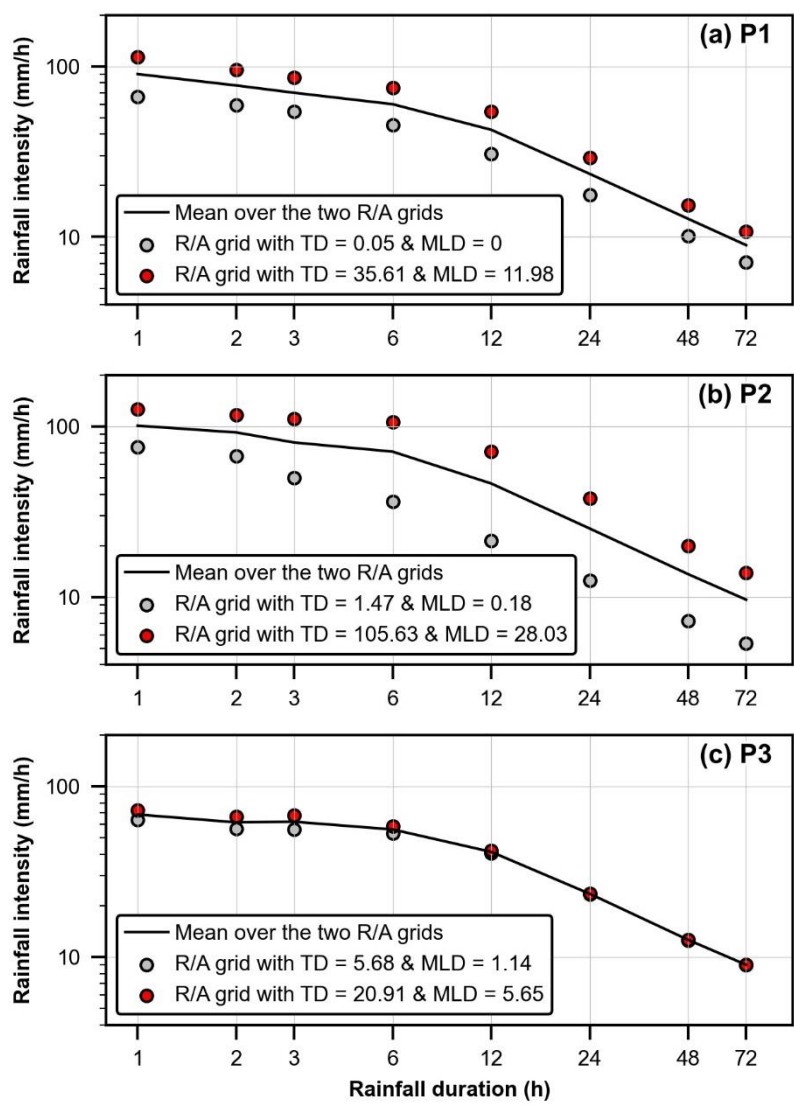

**Figure 4: Comparison of rainfall intensity maxima in three pairs of R/A grid cells with comparable local slope distributions within $A_{threshold}$.**

## 3.2 Relationship between landslide density and return levels of rainfall intensity maxima

During the examined rainfall event, the spatial patterns of rainfall return levels can be constraints for the variation of landslide density. The Gumbel distributions estimating these return levels were able to represent the observed AMS of rainfall intensities for 1–72 timespans, as the KS test could not reject the null hypothesis (*p-value* > 0.05) (Fig. S4). The rainfall intensities estimated for various return periods (5–100 years) and durations (1–72 h) displayed substantial spatial differences at the R/A grid cell scale (Figs. S5–S9). The Mann-Kendall and Sen's slope tests showed a spatial heterogeneity in the significance and magnitude of trends in observed rainfall AMS (Figs. S10 and 11). Specifically, some R/A grid cells in the western part of the study area showed statistically significant positive rainfall trends at the 95 % significance level, as the Mann-Kendall rejected the null hypothesis (*p-value* < 0.05). Other R/A grid cells exhibited no significant trends, especially for short-duration rainfall intensities (Fig. S10a–c), where Mann-Kendall accepted the null hypothesis (*p-value* > 0.05). The increasing trends could be attributed to the climate change effect and indicated that the rainfall IDF curves developed for the examined region are already subject to climate change and may be altered in the future due to the persistent effect of climate change. Still, they could provide valuable information about the return levels of the rainfall intensity maxima characterizing the examined rainfall event.

Comparing the position of rainfall intensity maxima in the IDF curves recorded for each R/A grid cell discloses disparate return levels (Figs. 5 and S12). The return levels of rainfall intensity maxima over the R/A grid cells with high landslide density metrics in the three idealized pairs (Fig. 5d–f) were generally higher than those observed over the corresponding R/A grid cells with lower landslide density metrics (Fig. 5a–c). In P1 and P2, rainfall return levels of all maxima over the high landslide density R/A grid cells (Fig. 5d and e) exceeded or hit the IDF curve for the 100-year return period. On the other hand, the return levels of rainfall intensity maxima exceeded the 100-year return period only at 6 and 12 timespans (Fig. 5a) and did not reach this level at any of the examined timespans (Fig. 5b) for the R/A grid cells with low landslide density in P1 and P2, respectively. Therefore, the number of triggered landslides increased substantially when rainfall return levels exceeded the 100-year return period in the IDF curves for the multiple examined timespans (i.e., 1–72 h).

Interestingly, despite the comparable rainfall intensities and slope distributions within the R/A grid cells in P3 (Fig. 4c), return levels of short-duration rainfall intensity maxima differed, as for the landslide density metrics (Fig. 5c and f). The return levels of rainfall intensity maxima in both R/A grid cells exceeded the 100-year return periods only for some timespans and shared comparable return levels for the rainfall intensity maxima at 12–72 h. Still, the rainfall return levels for 1–6 h-intensities in the high landslide density R/A grid cell (Fig. 5f) were higher than those observed in the R/A grid cells with lower landslide density (Fig. 5c). For instance, the return level of 3-h rainfall intensity exceeded the 100-year return period in the R/A grid cell with TD = 20.91 landslides/km$^2$ (Fig. 5f), but it was in the order of 50-year return period in the R/A grid cell with TD = 5.68 landslides/km$^2$ (Fig. 5c). Therefore, the results in P3 showed that the landslide density metrics over an R/A grid cell increased with the increase in rainfall return levels, rather than rainfall intensities.

The observations over the three idealized pairs showed that the spatial patterns of rainfall return levels constrain the variation of landslide density metrics observed during the examined event. For setting a quantitative reference that assesses the spatial disparity in rainfall return levels and their relation to the variation in landslide density, we calculated the ratio between the rainfall intensity maxima within the $P_{std}$ and the estimated rainfall intensity for a 100-year return period derived from the IDF curves. This index was referred to hereafter as the "100-year rainfall anomaly" and serves as a comparative index of the severity and rarity of rainfall intensity maxima observed over the R/A grid cells.

Clearly, the 100-year rainfall anomaly in the R/A grid cells with high landslide density was higher than that observed over the paired low landslide-density R/A grid cells in the idealized pairs (Fig. 5g–i). In P1 and P2, the 100-year rainfall anomaly exceeded one at all timespans in the case of the R/A grid cells with high landslide density, mirroring unprecedented and severe rainfall intensities. On the other hand, it was lower than or exceeded one only at some timespans for the R/A grid cells with lower landslide density (Fig 5 g, and h). In P3, the 100-year rainfall anomalies for 12–72 h rainfall durations observed over the two paired R/A grid cells were comparable. However, the 100-year rainfall anomalies for 1–6 h timespans were higher in the high landslide density R/A grid cell (Fig 5i), particularly for the 3-h rainfall duration, which exceeded one. Therefore, the comparison of the 100-year rainfall anomaly can indirectly reflect the difference in rainfall return levels and explain the spatial variation in landslide density observed over the R/A grid cells in the idealized pairs.

Irrespective of the differences in local slope distributions and rainfall characteristics between the R/A grid cells in the idealized pairs, landslide density metrics increased with the increase in the 100-year rainfall anomaly, except for the low landslide density R/A grid cells in P2 (Fig. 5h). For instance, the low landslide R/A grid cell in P1 (i.e., TD = 0.05 landslides/km²) and P3 (i.e., TD = 5.68 landslides/km²) showed different landslide density metrics. In parallel, the rainfall anomaly in the R/A grid cell with a TD = 5.68 landslides/km² was higher than that observed over the R/A grid cell with a TD = 0.05 landslides/km². Thus, comparing the 100-year rainfall anomaly may explain the spatial variation in landslide density observed in some of the R/A grid cells, irrespective of the differences in local slope distributions.

In this sense, we can categorize the R/A grid cells that experienced landslides (except three R/A grid cells where landslides were affected by anthropogenic activities) based on differences in the 100-year rainfall anomaly and landslide density. Accordingly, the high landslide density R/A grid cells (TD > 30 and MLD > 10 landslides/km²), of which the R/A grid cells with high landslide density in P1 and P2 showed a 100-year rainfall anomaly exceeded one at all timespans (Fig S13b). In other words, rainfall intensities for all examined timespans (i.e., 1–72 h) exhibited return levels exceeding the 100-year return period. While over lower landslide density R/A grid cells (TD < 30 and MLD < 10 landslides/km²), which include the R/A grid cells with low landslide density in P1 and P2 and the two paired R/A grid cells in P3, the 100-year rainfall anomaly was generally lower than one or exceeded one only at some timespans within the $P_{std}$ (Fig S13a).

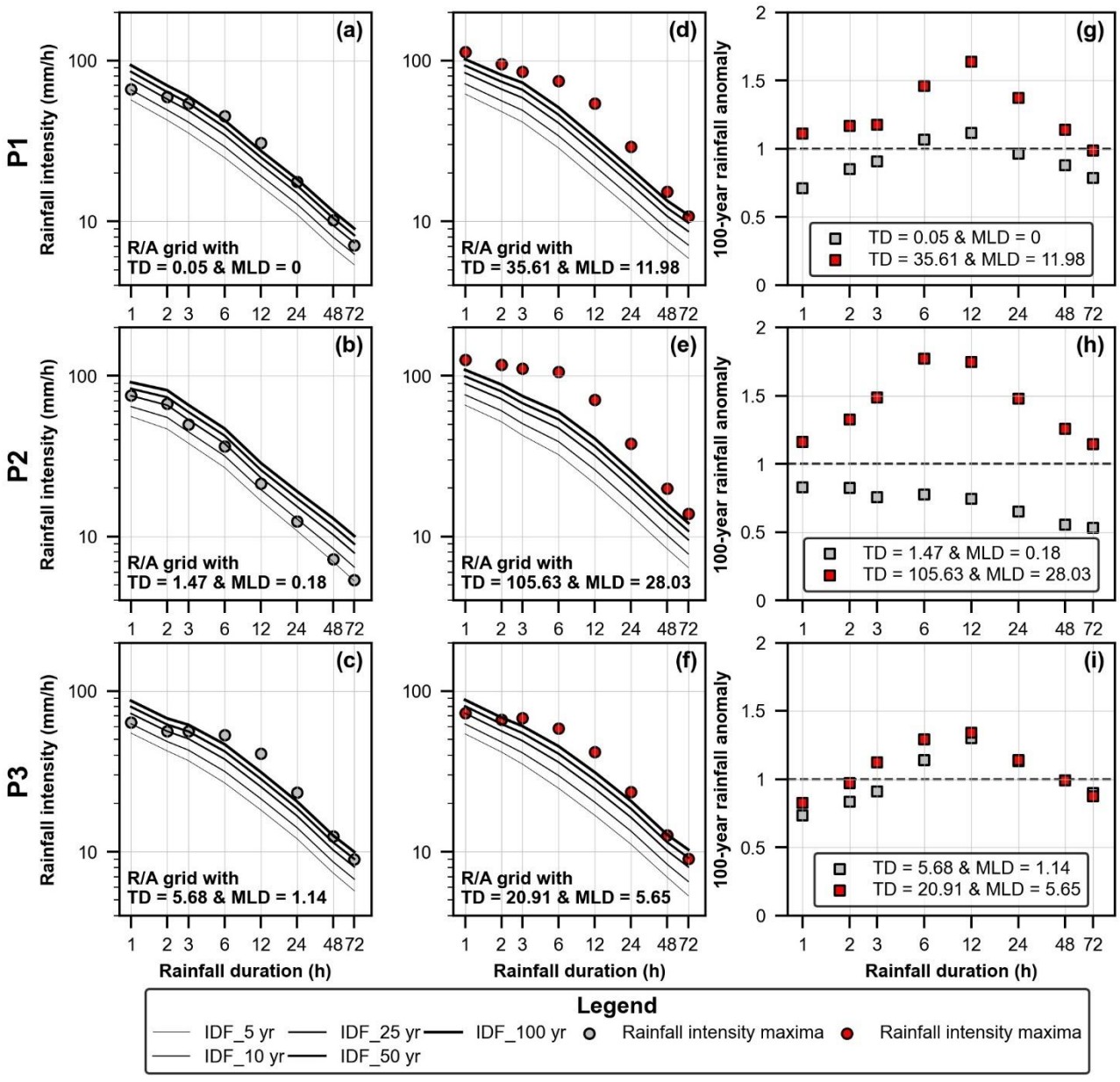

**Figure 5: Return levels of rainfall intensity maxima for multiple timespans (1–72 h) within $P_{std}$ in the IDF curves (a–f) and comparisons of the 100-year rainfall anomaly (g–i) over the paired R/A grid cells in P1, P2, and P3**

## 4 Discussion

### 4.1. Rainfall return levels govern landslide density

Our results demonstrate that landslide density in terms of TD and MLD varied depending on rainfall return levels for the examined timespans ranging from 1 to 72 h, which characterize the spatiotemporal rainfall pattern of the triggering rainfall event and provide proxies for the disparate rainfall periods needed for landsliding.

When rainfall exhibited return levels exceeding the 100-year return period for the various timespans from 1 to 72 hours (e.g., Fig. 5d, e), the number of total landsliding was substantially high (TD > 30 landslides/km$^2$). The high landslide density can dictate that the rare and extreme rainfall intensities for multiple timespans from 1 to 72 h could satisfy the trigger and dynamic predisposition factors for the landsliding of numerous hillslopes. The constraint of these unprecedented rainfall intensities on landslide density overwhelmed that of topographic conditions (Fig 5), as we observed substantial landslide density differences over R/A grid cells with comparable local slope distributions. This accentuates the importance of high rainfall return levels in inducing widespread landslides (Iida, 2004; Griffiths et al., 2009; Segoni et al., 2014). In parallel, the density of large and medium landslides was also the highest (MLD > 10 landslides/km$^2$) during the examined rainfall event. This implies that the high rainfall return levels for the various examined timespans constrain the occurrence of relatively large landslides and suggests that the spatiotemporal rainfall pattern characteristics can also govern the landslide size distribution, which is consistent with the findings of Marc et al. (2018). In contrast, when rainfall return levels did reach the 100-year return period only at specific timespans, lower landslide density (TD < 30 and MLD < 10 landslides/km$^2$) was observed (e.g., Fig. 5a, c, f). In other words, only some periods of rainfall (e.g., 6–48 h) exhibited extreme and rarely experienced intensities over the R/A grid cells, resulting in the failure of only the relatively vulnerable hillslopes. Therefore, we can conclude that whether rainfall intensities reach high return levels in a wide timespan, ranging from a few hours to several days, is one of the key determinants of the density of total landsliding and relatively large landslides.

Given the relatively homogeneous regolith of the study area this research focused on, it is likely that the landslide spatial distribution was primarily governed by rainfall return levels. However, other landslide susceptibility factors may intervene if the studied rainfall event is experienced in a heterogeneous regolith. To examine the importance of rainfall controls on landslide spatial distribution during large-scale rainfall events, Crozier (2017) proposed a storm cell model linking landslide density to rainfall intensity, impact magnitude, and the criticality of landslide susceptibility parameters. The proposed model assumes the occurrence of landslides in a circular pattern mirroring rainfall intensity during rainfall events and defines three landslide response zones: the core (storm center), the middle, and the periphery zone. It further suggests an overwhelm of the influence of extremely intense rainfall in the core zone, where total rainfall > 500 mm, on other landslide susceptibility factors.

In analogy to the storm cell model of Crozier (2017), the high rainfall return levels experienced over high landslide density grid cells may outweigh the influence of terrain-related parameters if experienced in other sites with heterogeneous regolith settings. Therefore, when rainfall intensities reach high return levels for a wide timespan ranging from an hour to a few days, high landslide density over the landscape can be expected regardless of the variations in terrain characteristics (land use, lithology, and topography). In contrast, when rainfall return intensities exceed the 100-year return level only for specific timespans (e.g., 6–48 h), the variation in landslide susceptibility factors can also govern landslide density. This can be supported in analogy to the findings of Crozier (2017) in the middle zone of the proposed storm model.

Last, it is worth noting that landslides occurred even when rainfall did not reach the 100-year return level at any of the examined timespans (Fig S12 b, e, f). However, landslide density over these grid cells (i.e., grid cells where rainfall did not reach the 100-year return level) was considerably low ($\approx 0.4$–$1.5$ landslides/km$^2$ in terms of TD) compared with most other grid cells. Dou et al. (2020) and Ozturk et al. (2021) used statistical machine-learning methods to investigate the importance of numerous predisposing factors in landslide occurrence by the examined rainfall event. Their findings showed that rainfall is the main factor controlling landslide occurrence in our study area, followed by the slope and land use parameters. Accordingly, landslide occurrence over these grid cells during the examined rainfall event could be constrained by terrain settings (e.g., land cover) as the rainfall return levels were low. Therefore, landslides can occur even if rainfall return levels do not reach the 100-year return period but with substantially low density. In any case, comparing rainfall return levels in the IDF curves can explain the substantial differences in landslide density due to considering multiple return periods.

## 4.2. Importance of considering rainfall return levels as explanatory for landslide spatial distribution

From a statistical perspective, the significant quantitative correlations between rainfall intensity maxima and landslide density (TD and MLD) suggest an increased landslide density with increased rainfall intensities for the various examined timespans (i.e., 1–72 h) (Table 1). These statistical relationships are not surprising since they likely arise from the correlations between the different rainfall intensity maxima (Table S2). However, this does not necessarily mean that landslide density increases with increased specific-duration rainfall intensity (e.g., rainfall intensity maxima for 6 h, Fig. 4a, c). Indeed, our results showed substantial differences in landslide density over R/A grid cells with comparable short-duration rainfall intensity maxima but disparate long-duration rainfall intensities (e.g., low landslide-density R/A grid cells in P1 and P3, Fig. 4a, c). The pronounced difference in landslide density is likely due to the disparity in rainfall characteristics that affected the slope stability differently, initiating a disparate number of landslides. Thus, although the quantitative correlations in Table 1 can successfully predict landslide density, as indicated by Chang et al. (2008) and Dai and Lee. (2001), relying on a single rainfall metric (e.g., 6 h rainfall intensity maxima) may lead to spurious interpretations regarding rainfall controls on landslide density and subject to uncertainties if used for predicting the number of landslides due to concealing the characteristics of the temporal rainfall pattern (Gao et al., 2018).

Regardless of the spatial variation in rainfall intensity maxima characterizing the temporal rainfall pattern, the return levels
could evaluate the exceptionality and extremity of rainfall for various timespans. Indeed, by comparing the rainfall return
levels over two R/A grid cells, it was clear that the R/A grid cells with the highest landslide density experienced higher
rainfall return levels for the various timespans, as revealed by the proposed 100-year rainfall anomaly metric (e.g., Fig. 5g–
i). This can dictate that rainfall with higher return levels was more extreme and less frequent, having a higher potential to
cause numerous landslides over the landscape. This was also valid even for R/A grid cells with comparable rainfall
intensities and local slope distributions emphasizing the constraint of rainfall return levels on landsliding rather than rainfall
intensities (Fig 5i). Accordingly, the differences in rainfall return levels could explain the substantial spatial disparity in
landslide density. Thus, the comparison of rainfall return levels can be a valid approach for understanding the substantial
differences in landslide density regardless of the variation in temporal rainfall pattern characteristics.

## 5 Conclusions

This study explored the spatiotemporal pattern of an extreme rainfall event that triggered widespread landslides to reveal
what rainfall characteristics control the spatial landslide distribution. We examined the temporal rainfall pattern by
computing the maximum rainfall intensity for multiple timespans (1–72 h) within a 72-h duration that accumulated the
maximum rainfall amount ($P_{std}$) during the examined rainfall event. Landslide density, in terms of the total number of
triggered landslides (TD) and only medium and large landslides (MLD), significantly correlated with all computed rainfall
intensity maxima. However, this did not necessarily mean that landslide density increases with increased rainfall intensity
maxima for a specific time span. More than 65 % of triggered landslides occurred in areas where all computed rainfall
intensity maxima exceeded or hit the 100-year return levels, with a high density (TD > 30 landslides/km$^2$ and MLD > 10
landslides/km$^2$). This corresponds to a 100-year rainfall anomaly, which calculates the ratio between rainfall intensity
maxima and estimated intensity for the 100-year return period, exceeding one at all timespans within the $P_{std}$. On the other
hand, lower landslide density was found in areas of rainfall characterized by intensities that did not or did reach the 100-
year return period only at some timespans within the $P_{std}$ (e.g., 6–48 h). The constraint of rainfall return levels on landslide
density overwhelmed that of topographic conditions, as we observed substantially different landslide densities in areas with
comparable slope distributions but different rainfall return levels. Overall, this work reveals the role played by the spatial
patterns of rainfall return levels for various timespans in controlling landslide density. It further suggests that whether
rainfall intensities reach high return levels for a wide timespan, ranging from a few hours to several days, is one of the key
determinants of the density of total landsliding and relatively large landslides.

## Code availability

The multidimensional analysis carried out in this paper used python open-source libraries: "rasterio" (Gillies and Others, 2013), "xarray" (Hoyer et al., 2021),"rioxarray" (Snow et al., 2021), and "xclim" (Logan et al., 2021). All figures were
created using the python open-source library Matplotlib (Caswell et al., 2021).

## Data availability

The landslide inventory data is available upon agreement by the Ministry of Land, Infrastructure, Transport, and Tourism of Japan (http://www.qsr.mlit.go.jp/), which holds the data copyright. The R/A precipitation data is a commercial product of the Japan Meteorological Agency and can be purchased from the Japan Meteorological Business Support Center
(http://www.jmbsc.or.jp/jp/). The DEM data used in this research can be freely downloaded from the GSI website (https://fgd.gsi.go.jp/download/menu.php).

## Competing interests

The contact author has declared that none of the authors has any competing interests.

## Acknowledgements

We acknowledge the Geospatial Information Authority of Japan (GSI) for freely providing the DEM data. Furthermore, we thank the anonymous referees for their constructive comments that greatly improved the quality of this paper.

## Financial support

This work was supported by the project "Development of Technology for Impacts, Mitigation and Adaptation to Climate Change in The Sectors of Agriculture, Forestry and Fisheries" of the Agriculture, Forestry, and Fisheries Research Council
(Japan).

## Author contributions

SM designed the study, performed the analyses, and wrote the paper. HT performed the FAD analysis and reviewed the paper.

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
