# Peer review of "Reveal the relation between spatial patterns of rainfall return levels and landslide density"

_Earth Surface Dynamics, 2022_

## Author Comment (AC1)

We thank the Associate Editor and three anonymous Referees for handling and assessing our manuscript. We are also grateful to the three referees for their insightful observations and critiques. Following their recommendations and concerns, we will carefully revise our manuscript to clarify the methods and significance of this research.

Hereafter, we provide preliminary responses to these observations before addressing them in detail in a revised manuscript.

The comments of the Associate Editor and the three Referees are in *italic black* font style. Our preliminary responses are in regular blue font style.

**Responses to the Associate Editor (EC1)**

**EC1: Comments and responses**

*Dear Authors,*
*We have now received three referee comments (RCs). Based on the RCs, major revisions may be needed before the manuscript may be considered for publication.*
*Please respond to the three Referee Comments. RC2, in particular, provided detailed critiques and suggestions for improving the manuscript.*
*Upload a revised manuscript and a detailed response to the RCs by March 10, 2023.*
*Best,*
*Sagy Cohen, Associate Editor*

Thank you again for handling our manuscript. We are grateful to the three referees for their insightful comments and critiques. We understand and respect the critiques given by RC2. However, we do feel that most of them originated from an intrinsic misunderstanding of our study hypothesis and methods, which might be due to an unclear explanation in the original manuscript. Hereafter, we provide preliminary responses to all observations of the referees to explain how we will revise our manuscript and clear out any potential misunderstandings.

We will carefully revise our manuscript following the recommendations of the three referees and upload a revised manuscript and detailed responses by the due date.

**Responses to Referee 1 (RC1)**

**RC1: Comment 1 and response**

*In the introduction part, the authors should clearly indicate the research gap and the novelty of this research.*

Thank you again for assessing our manuscript. We understand your concerns regarding the clarity of the research gap and novelty in our manuscript, which can be originated from an unclear explanation in the Introduction section. We will substantially revise the Introduction section in a revised manuscript to improve the presentation of our research gap and novelty.

**RC1: Comment 2 and response**

*The research object of this paper is mainly shallow landslides. It is recommended to highlight the uniqueness of the research object in the abstract and introduction.*

We acknowledge that most landslides triggered during the examined rainfall event are shallow (depth = 1 to 2 m), as indicated by Chigira et al. (2018). Still, some of the landslides could be relatively deep, as we observed a few landslides with large areas (area > 10,000 $m^2$) in the FAD of the landslide inventory (Figure 2 of the original manuscript).

Because we did not use any fundamental criteria to differentiate shallow landslides (e.g., area < 10,000 $m^2$ in Marc et al. (2019)) due to the unavailability of validation data (i.e., high-resolution DEM data taken before and after the examined event), we believe that adding "shallow landslides" may cause some confusion for readers. Therefore, we prefer not to limit the study to shallow landslides.

**RC1: Comment 3 and response**

*In Figure 1b, the north arrow is missing.*

We will add the missed north arrow in Figure 1b. Also, we will add the missed label and unit in the color bar of Figure 1a.

**RC1: Comment 4 and response**

*In Figure 3, the contour of the study area should be added. The color bars in Figure 3 lack labels and units. Please check similar issues in other figures.*

We have added the contour lines of the study area in Figure 3 (please see Fig. RC1.1). However, we feel that the figure becomes unclear for readers as it overlays multiple different information (i.e., rainfall intensity, landslide distribution, TD, MLD, and contour lines). We believe that adding the contour lines may make the figure difficult to understand. Therefore, we prefer not to add it. On the other hand, we will add the missed labels and units in the color bars in Figure 3 and all other figures in the Supplement file.

**RC1: Comment 5 and response**

*The discussion part needs to be reorganized.*

We will reorganize the discussion section in a revised manuscript with sub-sections to make it more accessible for readers.

**RC1: Comment 6 and response**

*Figure 4c: "TD 5.68 & MLD = 1.14" should be changed to "TD = 5.68 & MLD = 1.14".*

We will re-create Figure 4 to correct this mistake.

**RC1: Comment 7 and response**

*Line 255: The 100-year rainfall anomaly was higher in the low landslide-density grid cell in P3 (Fig. 5i) than in the low landslide-density grid cell in P1 (Fig. 5c) (< 1.5 times). Why could the comparison of the 100-year rainfall anomaly explain the substantial difference in landslide density between the two grid cells (≈ 110 times for TD).*

This important question leads us to notice an insufficient explanation regarding the use of the 100-year rainfall anomaly in our study. Therefore, we will substantially clarify it in our revised manuscript.

It is worth noting that the 100-year rainfall anomaly was proposed in our study to assess the rainfall intensity for multiple timespans (i.e., rainfall intensity maxima) in terms of rarity and extremity rather than rainfall intensity. For instance, a 100-year rainfall anomaly for a 3-h timespan higher than 1 means that the 3-h maximum rainfall intensity was extreme and rare compared to previously experienced 3-h rainfall intensity in the study area as it has a return level of > 100-year return period. Thus, the 100-year rainfall anomaly can provide important information on the potential of the multiple rainfall timespans to induce landslides, as high return level rainfall is generally needed for landsliding (Iida, 1999; Segoni et al., 2015). Accordingly, it can be a standard method to compare the potential of rainfall intensity maxima observed in the different R/A grid cells to trigger landsliding, irrespective of the differences in rainfall intensity maxima.

We found that the 100-year rainfall anomaly was higher in the low landslide-density grid cell in P3 (Fig. 5i) than in the low landslide-density grid cell in P1 (Fig. 5c). This means that rainfall timespans in the former were more extreme (i.e., high potential to cause landslides) than those experienced over the latter. Accordingly, the differences in the 100-year rainfall anomaly, which dictate the potential of rainfall periods to cause landsliding, could explain the substantial difference in landslide density over the two R/A grid cells.

[Figure]

**Figure RC1.1.** Spatial distribution maps of rainfall intensity maxima for 1 to 72 h timespans within the $P_{std}$ in mm/h, triggered landslides (grey polygons), and landslide density metrics (circles). The brown lines show the contour lines of the study area.

**References**

Chigira, M., Sixian, L., and Matsushi, Y.: Landslide disaster induced by the 2017 northern Kyushu rainstorm, Disaster Prevention Research Institute Annals, 28-35 (in Japanese, with English abstract) pp., 2018.

Iida, T.: A stochastic hydro-geomorphological model for shallow landsliding due to rainstorm, Catena, 34, 293–313, https://doi.org/10.1016/S0341-8162(98)00093-9, 1999.

Marc, O., Gosset, M., Saito, H., Uchida, T., and Malet, J. P.: Spatial Patterns of Storm-Induced Landslides and Their Relation to Rainfall Anomaly Maps, Geophys. Res. Lett., 46, 11167–11177, https://doi.org/10.1029/2019GL083173, 2019.

Segoni, S., Battistini, A., Rossi, G., Rosi, A., Lagomarsino, D., Catani, F., Moretti, S., and Casagli, N.: Technical Note: An operational landslide early warning system at regional scale based on space-time-variable rainfall thresholds, Nat. Hazards Earth Syst. Sci., 15, 853–861, https://doi.org/10.5194/nhess-15-853-2015, 2015.

**Responses to Referee 2 (RC2)**

**RC2: Comment 1 and response**

*The study relates a large data set of landslides with rainfall characteristics in Japan, using 7,500 landslides over an area of 400km². The study uses radar precipitation at 25km2 resolution with 1 to 72 h durations. Land cover and lithology are deemed homogenous in the study site.*
*A power-law distribution is used to identify the landslide size cutoff for moderate and large sizes. Landslide densities are only calculated where slopes exceeded a threshold of 16.26 degrees (slopes that include >90% of slides). Landslides are separated into total landslide density (TD), which includes all the observations, and medium and large landslide size density (MLD), which includes the slides greater that the size cutoff (>439 m2).*
*A standardized rainfall that accumulates maximum rainfall over 72h period is used as Pstd. Within this Pstd, multiple time periods that record maximum intensities were also identified (1h to 72h). That aided the authors to develop a rainfall intensity-duration relation threshold curves based on I-D data.*
*Figure 3 presents a map of 1h to 72h maximum rainfall depths (25km2 resolution) along with TD and MLDs. Higher landslide densities are observed where rainfall intensities are high.*
*More landslides occurred with rainfall exceeded 100 year return interval.*

Thank you for assessing our manuscript. We like to clarify a potential misunderstanding about how we calculated landslide density in this study. Our study intended to examine the interplay between rainfall intensity for multiple timespans, which can be assessed by their return levels, and the spatial variation of landslide density. Given that the rainfall information was derived from a 5-km radar-driven gauge-adjusted precipitation dataset (referred to as R/A), we calculated landslide density by considering the number of landslides that occurred within each R/A grid cell. This is different from other studies that intended to examine how landslide density varies with slope angle, and therefore they calculated landslide density by counting the number of landslides that occurred within particular ranges of local hillslope angles (Coe et al., 2004; De Rose, 2013; Prancevic et al., 2020).

So, differently from what is stated, "*Landslide densities are only calculated where slopes exceeded a threshold of 16.26 degrees (slopes that include >90% of slides)*", landslide densities considered the number of **all** landslides (for total landslide density "TD") and **all** landslides with area > 439 m² (for medium and large landslides density "MLD") occurred within each R/A grid cell (i.e., ≈ 25 km²). The threshold of 16.26° (considered in our study as a minimum slope threshold to allow landsliding and referred to as $S_{threshold}$) was used to calculate the area of the R/A grid cells where the slope > 16.26° (referred to hereafter as $A_{S>16.26°}$). The two Landslide density metrics were, therefore, calculated by dividing the number of landslides (i.e., **all** landslides for TD and **all** landslides with an area > 439 m² for MLD) that occurred within each R/A grid cell by $A_{S>16.26°}$ following the equation (1) and (2).

$$\text{TD} = \frac{\text{Total number of all landslides within the R/A grid cell}}{A_{S>16.26°}} \qquad (1)$$

$$\text{MLD} = \frac{\text{Number of medium and large landslides within the R/A grid cell}}{A_{S>16.26°}} \qquad (2)$$

Such a normalization method is fundamental to reduce bias in the numbers of triggered landslides within the different R/A grid cells caused by the differences in the distribution of local topographic features (Prancevic et al., 2020), as landslides commonly occur in hilly and mountainous areas rather than plains (Lombardo et al., 2021). Therefore, it makes assessing the relationship between rainfall information and landslide densities in the R/A grid cells less biased by the differences in local slope conditions. We note that such a normalization method has been also adopted in some previous works by considering a 10° slope as the minimum slope threshold for landsliding (Marc et al., 2019) or the slope at which > 90 % of landslides occurred (Prancevic et al., 2020).

**RC2: Comment 2 and response**

*Observations: P1, P2, P3-- can you clarify how the populations of landscape slopes similar in these groups, do you report any statistics somewhere? Where are those populations? Are they identified within each selected rainfall grid or can they be located in different rainfall grids?*

It is worth noting that each of the pairs (i.e., P1, P2, and P3) represents two R/A grid cells with similar local slope conditions within $A_{S>16.26°}$ but different landslide density metrics (i.e., TD and MLD). The selection of the three pairs was based on the distribution of local slope conditions in $A_{S>16.26°}$ of the different R/A grid cells rather than landslide data. In other words, we examined all slope pixels (resolution = 10 m) in $A_{S>16.26°}$ and did not limit the analysis to only landslide slope pixels. By selecting these pairs, we intended to explicitly focus on rainfall controls and avoid any possible influence of the differences in local slope conditions of $A_{S>16.26°}$ of the R/A grid cells on landslide occurrence.

The three pairs were selected by first comparing the distribution of slope conditions in $A_{S>16.26°}$ of all R/A grid cells (i.e., 23) using the Kruskal-Wallis static (Kruskal and Wallis, 1952) to validate the existence of significant differences in local slope conditions. To better highlight these differences, we provided a Figure showing the distribution of local slope degrees in $A_{S>16.26°}$ of the different R/A grid cells referred to in this figure by the corresponding TD (please see Fig. RC2.1). Subsequently, we employed Dunn's post hoc test for detecting the R/A grid cells with a similar mean rank sum of slopes, meaning similar slope conditions. We note that the result of Dunn's test has been already shown in Table S1 in the Supplement file, as stated in our preprint (P8, L198). From this result, we could find three pairs of R/A grid cells characterized by similar slope conditions (as Dunn's test could not reject the null hypothesis) and different landslide density metrics. Therefore, to explicitly reveal the controls of rainfall information on landslide density, we mainly focused on these three pairs (i.e., P1, P2, and P3) as each pair of R/A grid cells includes two R/A grid cells with similar local slope conditions.

[Figure]

**Figure RC2.1.** Distribution of local slope degree within $A_{S>16.26°}$ of the R/A grid cells. Note that the distributions are shown as box-and-whisker plots. The box delimitates the 25th and 75th percentiles. The black line indicates the median. The red cross '+' displays the mean. The circles 'o' designate the outliers.

**RC2: Comment 3 and response**

*Lines 195-220: I'm not sure what the objective here, if one is interested to find out where rainfall plays a stronger role, then shouldn't you go and investigate the local conditions (area, slope, soil veg properties) of individual slides.*

Here, we compared the relation between rainfall intensity maxima and landslide density in three pairs of R/A grid cells with similar slope conditions (i.e., P1, P2, and P3). We intended to explore the potential interplay between the rainfall intensity maxima and the spatial variation of landslide density metrics (i.e., TD and MLD).

We agree that one of the methods is to investigate local conditions (e.g., slope, soil, vegetation properties, etc.). However, there are mainly one or two controlling factors in some specific regions which are worth exploring. In our study area in particular, two interesting previous works have investigated the importance of multiple predisposing factors (e.g., slope, land cover, elevation) in landslide occurrence using statistical machine-learning methods (Ozturk et al., 2021; Dou et al., 2020). Both works showed that rainfall is the main factor controlling landslide occurrence in our study area, followed by the slope and land use parameters. These findings were also consistent with the in-field observation of Chigira et al. (2018). It is worth noting also that several previous works showed the feasibility to assess only rainfall conditions

for landslide prediction by exploring the spatial relation between rainfall conditions and landslide density (Chen et al., 2013; Chang et al., 2008; Dai and Lee, 2001; Gao et al., 2017; Marc et al., 2019), as rainfall is the main factor for landsliding. Given this, we mainly focused on rainfall controls on landslide density in this study.

*I think the selection process of P groups are based on some random selection routine, if you shuffle these landslides into another set of 3 populations you may get all three look like P1 and P2 with smaller differences in rainfall rate differences, then what would you do.?*

From this comment, we believe you interpreted the selection of the three pairs as it was based on a random selection from the landslide data. Very differently, the three selected pairs of R/A grid cells were selected based on local slope conditions in the R/A grid cells. Please see our response to your second comment (RC2: Comment 2 and response), where we have cleared out how we selected the three pairs of R/A grid cells.

We believe this potential misunderstanding might be originated from unclear explanation of the selection method and intention of the three pairs (P1, P2, and P3). Therefore, we will clarify this by substantially revising our manuscript (in particular, Section 2.3.)

*I also could not figure out what those two different groups are within each plot in Figure 4. Why do the gray symbols have smaller landslide densities than red symbols? I think those were referred to as "pairs" but not sure how paired and why with different densities? Beyond all what is the purpose of pairing.*

In Figure 4, each plot compared rainfall intensity for multiple timespans (i.e., rainfall intensity maxima) recorded in two R/A grid cells with similar slope conditions (for the $A_{S>16.26°}$), but different numbers of landslides as can be revealed by the two landslide density metrics (i.e., TD and MLD). For instance, in Fig. 4a, the gray symbols reflect the rainfall intensity maxima recorded in the R/A grid cell where TD = 0.05 and MLD = 0 landslides/km$^2$. The red dots reflect the rainfall intensity maxima observed in the R/A grid cell where TD = 35.61 and MLD = 11.98 landslides/km2. The black line showed the average rainfall intensity maxima in the two R/A grid cells in comparison.

The pairing approach we used in this paper aimed at selecting the R/A grid cells with similar slope conditions to avoid any possible influence of the differences in slope conditions on landslide density and explicitly focus on rainfall controls, as we explained in our response to your second comment (RC2: Comment 2 and response).

**RC2: Comment 4 and response**

*Rainfall data is very coarse for a rugged terrain to obtain any detailed and new science with respect to landslide process understanding and how rainfall controls it. The study may be useful for regional early warning systems, though still very coarse.*

We agree that high-resolution rainfall data would provide more detailed information on spatial rainfall patterns. However, long-term gridded rainfall data with a spatial resolution finer than 5 km, needed in our study to estimate rainfall return levels, is currently unavailable in Japan. Indeed, the R/A dataset used in this study is, so far, the highest-resolution and most reliable long-term gridded precipitation data available. Due to its relatively high resolution, long-term records, and accuracy, several studies used the R/A dataset as referential data for analyzing localized heavy rainfall (e.g., Kato, 2020; Hirockawa et al., 2020; Saito and Matsuyama, 2015), evaluating precipitation forecasts and estimates (e.g., Kubota et al., 2009; Iida et al., 2006; Yin et al., 2022), and constraining empirical relationships between rainfall information and landslide occurrence (e.g., Saito et al., 2010; Marc et al., 2019; Ozturk et al., 2021). All these works showed the usefulness of the R/A precipitation product in capturing the spatial pattern of extreme rainfall events experienced over the Japanese archipelago. Interestingly, Ozturk et al. (2021) evaluated the performance of a coarsened R/A dataset to ≈ 10-km resolution in landslide forecasting using a logistic regression model and showed a comparable performance between the 5-km and 10-km R/A dataset, meaning that the spatial rainfall pattern over the mountainous study areas Ozturk et al. (2021) focused on can be satisfactorily captured even with a 10-km spatial resolution R/A data. Therefore, as our objective was to assess the spatial relation between rainfall characteristics and landslide density, rather than explicitly examine the landsliding process of each of the triggered landslides, we believe that a resolution of 5 km could be sufficient due to its performance in capturing the spatial pattern of the studied rainfall event and given the unavailability of alternative product with finer resolution and long-term records.

*How do you take the next step from coarse-grain analysis to finer scale hazard mapping?*

We believe that the R/A data can be downscaled to finer resolution by employing machine learning and data fusion methods (e.g., Peleg et al., 2018; Salcedo-Sanz et al., 2020) to address finer scale hazard analysis. However, several drawbacks can limit the application of these methods, such as the need for dense rain gauges network over mountainous regions, which is generally difficult to obtain. We believe that rainfall data downscaling is another research issue that needs to be addressed in detail in the future and is beyond the objective of the current study.

**RC2: Comment 5 and response**

*What is the point of Figure 5, what is the question you are trying to address?*

It is worth recalling that all rainfall intensity maxima (i.e., maximum rainfall intensities for multiple timespans within the $P_{std}$) could explain the spatial variation of landslide density, as shown in Table 1 and Fig. 4. However, it is difficult to set a method to compare all rainfall intensity maxima between the different R/A grid cells that experienced landslides during the examined rainfall event. On the other hand, the return levels would assess the rainfall intensity maxima in terms of extremity and rarity comparing to rainfall intensity previously experienced in the R/A

grid cells. Accordingly, it can provide important information on the potential of these rainfall intensity maxima to induce landslides, as high return level rainfall is generally needed for landsliding (Iida, 1999; Segoni et al., 2015), irrespective of the rainfall intensity. Thus, rainfall return levels can be a standard method to compare the potential of rainfall intensity maxima to cause landslidiing at the spatial scale, irrespectively of the spatial disparity of rainfall intensity maxima of the examined rainfall event. Given this, In Figure 5, we compared the return levels of rainfall intensity maxima recorded over two R/A grid cells with similar local slope conditions and different landslide densities (i.e., P1, P2, and P3). Here, we intended to investigate whether the landslide density increases in the R/A grid cells where rainfall intensities reach high return levels that are rarely experienced.

*As far as I understood you have some randomly selected data pairs with different landslide densities and they seem to show some narrow range of variable ID trends, but this is expected isn't it.*

Sorry, you misunderstood how we selected the three pairs of R/A grid cells. The selection of these pairs was based on local slope conditions in the R/A grid cells rather than a random selection of landslide data. Please see our response to your second and third comments for more explanation (RC2: Comment 2 and response, RC2: Comment 3 and response).

*Another point I did not understand—in Figs 3 and 4, do each of the circles average many points with different landslide densities?*

Fig. 3 shows the spatial distribution of rainfall intensities for multiple timespans, triggered landslides, and landslide density metrics. Each white and black circle is the TD and MLD in the corresponding R/A grid cell, respectively.

No, in Fig. 4, each plot compared rainfall intensities for multiple timespans recoded in two R/A grid cells with similar slope conditions (for the $A_{S>16.26°}$), but different numbers of landslides as can be revealed by the two landslide density metrics (i.e., TD and MLD). So, the circles (red and gray) are the rainfall intensities for multiple timespans recorded in two R/A grid cells. For instance, in Fig. 4a, the gray symbols reflect the rainfall intensities for multiple timespans recorded in the R/A grid cell where TD = 0.05 and MLD = 0 landslides/km². The red dots reflect the rainfall intensities for multiple timespans recorded in the R/A grid cell where TD = 35.61 and MLD = 11.98 landslides/km². The black line showed the average of rainfall intensities between the two R/A grid cells in comparison.

**RC2: Comment 6 and response**

*Not having a clear research question and/or hypotheses makes it difficult to follow this paper.*

Our scientific question was to investigate the potential interplay between rainfall intensity for multiple timespans, which characterize the temporal rainfall pattern and can be assessed by

their return levels, and the spatial pattern of landslide distribution during the examined triggering rainfall event (i.e., landslide density spatial pattern). In other words, we intended to assess whether the spatial variation of landslide density during the examined triggering rainfall event is governed by the return levels of rainfall intensity for multiple timespans rather than rainfall intensity of a specific timespan (e.g., 48 h maximum rainfall intensity).

We understand your concern about the clarity of our research question and hypothesis. Therefore, we will substantially improve the introduction section in a revised manuscript to clearly state our research question and hypothesis.

*In addition, the methods rely on some comparisons of three similar slope populations (P1,2,3), and pairing of data among them, the purpose of which was not clear.*

Sorry, you misunderstood how and why we select the three pairs of R/A grid cells with similar slope conditions. Please see our response to your second and third comments for more explanation (RC2: Comment 2 and response, RC2: Comment 3 and response).

To clear this out and avoid any potential future misunderstanding, we will improve the Methods section in a revised manuscript.

*If the whole point of the paper is to show that rainfall patterns and return intervals matter, that is no surprise to anyone, that is why those intensity-duration thresholds were used for nearly a century.*

First, it is worth noting the existence of two empirical approaches for quantifying rainfall characteristics that triggered landslides. The first approach is the traditional intensity-duration (ID) thresholds that determined the minimum rainfall conditions necessary for likely triggering landslides. The second approach, mainly used in this paper, relates the spatial variation of landslide density with rainfall information beyond the ID thresholds.

The objective of this paper was to mainly investigate the potential interplay between rainfall intensity for multiple timespans, which characterize the temporal rainfall pattern and can be assessed by their return levels, and the spatial variation of landslide density. We showed that landslide density is constrained by the return levels of rainfall variables for multiple timespans rather than the intensity of a single rainfall timespan (e.g., Maximum rainfall intensity for 48 h). Our finding is different from other studies' findings that related the spatial variation of landslide density to a single rainfall variation for a specific timespan. Also, this is different from the ID thresholds that generally linked the occurrence of landslides to specific rainfall conditions in terms of intensity and duration. So, given this, we believe that the findings of our paper are novel and addressed a significant gap in the understanding of rainfall controls on landslide density.

*In addition, the rainfall data is at 5km spatial resolution, which for mountain ranges, is very coarse, and radar rainfall is usually not a good option for estimating mountain rainfall.*

We are aware of the intrinsic drawbacks of weather radars in reliably observing precipitation, which could be attributed to various meteorological, topographic, and technical factors (e.g., beam blockage, ground clutter, anomalous beam propagation, and range effects) (e.g., Borga et al., 2022). Therefore, we agree with the Referee's statement: "*radar rainfall is usually not a good option for estimating mountain rainfall.*" However, we believe this is the case for the raw uncorrected radar-driven precipitation data (e.g., Young et al., 1999). Differently, the R/A dataset used in this study was processed by a quality control algorithm involving various correction procedures for precipitation observation errors (Makihara, 2000; Hotta, 2018; Nagata, 2011). For instance, ground clutter and beam blockage due to mountains are corrected using a 2-km Pseudo Constant Altitude Plan Position Indicator (PCAPPI) that processes echo intensity data from multiple elevation angles. Additionally, the R/A product involves a Gauge-adjustment algorithm that calibrates precipitation estimates with gauge measurements. These correction procedures made the R/A product valuable for providing reliable rainfall estimates over the mountainous areas in Japan, which cannot be captured by rain gauged due to a sparse network. Therefore, it is often used as benchmark rainfall data in multiple studies over mountainous areas (please see RC2: Comment 4 and response).

It is worth noting, finally, that several previous studies showed the usefulness of corrected radar-driven precipitation datasets in observing the rainfall over mountains (e.g., Germann et al., 2006; Shimada et al., 2016; Nelson et al., 2016; Marra et al., 2022). Therefore, we believe that the R/A product used in our study provides reliable rainfall estimates over the mountainous areas in Japan.

*And finally, which is probably more important than any of the comments I made above, besides local slopes, the authors have not factored in elevation in their analysis. Elevation is also a good predictor of rainfall and variations in soils and vegetation. They used a slope threshold in their analysis to select landslides but a quick grouping by elevation would probably reveal a strong elevation control.*

It is worth recalling that the slope threshold (16.26°) was used only for deriving normalized landslide densities over the R/A grid cells while accounting for the number of **all** landslides (for TD) and **all** landslides with area > 439 $m^2$ (for MLD).

Of course, we agree that the elevation can have a strong control on landslide occurrence in addition to other predisposing factors for landslide occurrence (e.g., slope, land cover, rainfall, etc.). However, there are mainly one or two controlling factors in some specific regions which are worth exploring. For our study case in particular, Ozturk et al. (2021) evaluated the importance of multiple predisposing factors for landslide occurrence, including elevation and rainfall, using multivariate logistic regression. Their findings indicated that the rainfall information is the main

control for the spatial distribution of triggered landslides, followed by the slope parameter. On the other hand, the elevation parameter was found to be very less important in controlling landslide occurrence according to their findings.

To further assess how landslide occurrence varies with elevation, we have plotted the histograms of landslide elevations (i.e., 7,676 landslides) from a 10-m DEM (please see Figure RC2.2.). We found that the landslides occurred in hillslopes with a wide range of elevation from ≈ 50 to ≈ 800 m a.s.l. Although most of the landslides occurred in hillslopes with an elevation in the range of ≈ 50 to ≈ 600 m a.s.l., still, this elevation range is wide, meaning that landslide do not preferentially occurred on hillslopes with a specific elevation.

Given this, we believe that the elevation has a weak control on the spatial distribution of the landslides we focused on in this study. To avoid any similar queries by readers, we will add this information in the revised manuscript to clearly state the importance of rainfall controls in our examined study case.

[Figure]

**Figure RC2.2.** Non-cumulative (gray histogram) and cumulative (black line) frequency distribution of landslide elevations (bins = 500). Note that landslide elevations were calculated as the median of DEM pixel values at landslide scars.

*All in all, the paper left me with no new information. If the authors would want to salvage this paper, they would probably reconsider a set of new methods and pose clear questions and objectives.*

We respect your critiques. However, we feel that most of them originated from an intrinsic misunderstanding of the research methods, especially the method of landslide density calculation and pairs selection. Considering the research objective was to mainly investigate the potential interplay between a wide range of rainfall explanatory variables, which characterize the temporal rainfall pattern, and the spatial variation of landslide density, we believe that the methods used in our study could sufficiently address the research question.

Finally, we apologize for any misunderstandings which might be originated from unclear explanations of the research methods and hypothesis in the original manuscript. We will substantially improve the manuscript to clearly state our research questions and explain the methods.

**References**

Borga, M., Marra, F., and Gabella, M.: Rainfall estimation by weather radar, in: Rainfall: Modeling, Measurement and Applications, edited by: Renato, M., Elsevier, 109–134, https://doi.org/10.1016/b978-0-12-822544-8.00016-0, 2022.

Chang, K. T., Chiang, S. H., and Lei, F.: Analysing the Relationship Between Typhoon-Triggered Landslides and Critical Rainfall Conditions, Earth Surf. Process. Landforms, 33, 1261–1271, https://doi.org/10.1002/esp, 2008.

Chen, Y. C., Chang, K. T., Chiu, Y. J., Lau, S. M., and Lee, H. Y.: Quantifying rainfall controls on catchment-scale landslide erosion in Taiwan, Earth Surf. Process. Landforms, 38, 372–382, https://doi.org/10.1002/esp.3284, 2013.

Chigira, M., Sixian, L., and Matsushi, Y.: Landslide disaster induced by the 2017 northern Kyushu rainstorm, Disaster Prevention Research Institute Annals, 28-35 (in Japanese, with English abstract) pp., 2018.

Coe, J. A., Michael, J. A., Crovelli, R. A., Savage, W. Z., Laprade, W. T., and Nashem, W. D.: Probabilistic assessment of precipitation-triggered landslides using historical records of landslide occurence, Seattle, Washington, Environ. Eng. Geosci., 10, 103–122, https://doi.org/10.2113/10.2.103, 2004.

Dai, F. C. and Lee, C. F.: Frequency-volume relation and prediction of rainfall-induced landslides, Eng. Geol., 59, 253–266, https://doi.org/https://doi.org/10.1016/S0013-7952(00)00077-6, 2001.

Dou, J., Yunus, A. P., Bui, D. T., Merghadi, A., Sahana, M., Zhu, Z., Chen, C. W., Han, Z., and Pham, B. T.: Improved landslide assessment using support vector machine with bagging, boosting, and stacking ensemble machine learning framework in a mountainous watershed, Japan, Landslides, 17, 641–658, https://doi.org/10.1007/s10346-019-01286-5, 2020.

Gao, Z., Long, D., Tang, G., Zeng, C., Huang, J., and Hong, Y.: Assessing the potential of satellite-based precipitation estimates for flood frequency analysis in ungauged or poorly gauged tributaries of China's Yangtze River basin, J. Hydrol., 550, 478–496, https://doi.org/10.1016/j.jhydrol.2017.05.025, 2017.

Germann, U., Galli, G., Boscacci, M., and Bolliger, M.: Radar precipitation measurement in a mountainous region, Q. J. R. Meteorol. Soc., 132, 1669–1692, https://doi.org/10.1256/qj.05.190, 2006.

Hirockawa, Y., Kato, T., Tsuguti, H., and Seino, N.: Identification and classification of heavy rainfall areas and their characteristic features in Japan, J. Meteorol. Soc. Japan, 98, 835–857, https://doi.org/10.2151/jmsj.2020-043, 2020.

Hotta, J.: Hands-on Training on Weather Radar QC, in: WMO/ASEAN Training Workshop on Weather Radar Data Quality and Standardization Hands-on, 2018.

Iida, T.: A stochastic hydro-geomorphological model for shallow landsliding due to rainstorm, Catena, 34, 293–313, https://doi.org/10.1016/S0341-8162(98)00093-9, 1999.

Iida, Y., Okamoto, K., Ushio, T., and Oki, R.: Simulation of sampling error of average rainfall rates in space and time by five satellites using radar-AMeDAS composites, Geophys. Res. Lett., 33, 1–4, https://doi.org/10.1029/2005GL024910, 2006.

Kato, T.: Quasi-stationary band-shaped precipitation systems, named "senjo-kousuitai", causing localized heavy rainfall in japan, J. Meteorol. Soc. Japan, 98, 485–509, https://doi.org/10.2151/jmsj.2020-029, 2020.

Kruskal, W. H. and Wallis, W. A.: Use of Ranks in One-Criterion Variance Analysis, J. Am. Stat. Assoc., 47, 583–621, https://doi.org/10.1080/01621459.1952.10483441, 1952.

Kubota, T., Ushio, T., Shige, S., Kida, S., Kachi, M., and Okamoto, K.: Verification of high-resolution satellite-based rainfall estimates around japan using a gauge-calibrated ground-radar dataset, J. Meteorol. Soc. Japan, 87 A, 203–222, https://doi.org/10.2151/jmsj.87a.203, 2009.

Lombardo, L., Tanyas, H., Huser, R., Guzzetti, F., and Castro-Camilo, D.: Landslide size matters: A new data-driven, spatial prototype, Eng. Geol., 293, https://doi.org/10.1016/j.enggeo.2021.106288, 2021.

Makihara, Y.: Algorithms for precipitation nowcasting focused on detailed analysis using radar and raingauge data, Technical Reports of the Meteorological Research Institue, 63–111 pp., 2000.

Marc, O., Gosset, M., Saito, H., Uchida, T., and Malet, J. P.: Spatial Patterns of Storm-Induced Landslides and Their Relation to Rainfall Anomaly Maps, Geophys. Res. Lett., 46, 11167–11177, https://doi.org/10.1029/2019GL083173, 2019.

Marra, F., Armon, M., and Morin, E.: Coastal and orographic effects on extreme precipitation revealed by weather radar observations, Hydrol. Earth Syst. Sci., 26, 1439–1458, https://doi.org/10.5194/hess-26-1439-2022, 2022.

Nagata, K.: Quantitative Precipitation Estimation and Quantitative Precipitation Forecasting by the Japan Meteorological Agency, RSMC Tokyo–Typhoon Center Technical Review, 37–50 pp., https://doi.org/Online at: http://www.jma.go.jp/jma/jma-eng/jma-center/rsmc-hp-pub-eg/techrev/text13-2.pdf, 2011.

Nelson, B. R., Prat, O. P., Seo, D. J., and Habib, E.: Assessment and implications of NCEP stage IV quantitative precipitation estimates for product intercomparisons, Weather Forecast., 31, 371–394, https://doi.org/10.1175/WAF-D-14-00112.1, 2016.

Ozturk, U., Saito, H., Matsushi, Y., Crisologo, I., and Schwanghart, W.: Can global rainfall estimates (satellite and reanalysis) aid landslide hindcasting?, Landslides, 18, 3119–3133, https://doi.org/10.1007/s10346-021-01689-3, 2021.

Peleg, N., Marra, F., Fatichi, S., Paschalis, A., Molnar, P., and Burlando, P.: Spatial variability of extreme rainfall at radar subpixel scale, J. Hydrol., 556, 922–933, https://doi.org/10.1016/j.jhydrol.2016.05.033, 2018.

Prancevic, J. P., Lamb, M. P., McArdell, B. W., Rickli, C., and Kirchner, J. W.: Decreasing Landslide Erosion on Steeper Slopes in Soil-Mantled Landscapes, Geophys. Res. Lett., 47, 1–9, https://doi.org/10.1029/2020GL087505, 2020.

De Rose, R. C.: Slope control on the frequency distribution of shallow landslides and associated soil properties, North Island, New Zealand, Earth Surf. Process. Landforms, 38, 356–371, https://doi.org/10.1002/esp.3283, 2013.

Saito, H. and Matsuyama, H.: Probable hourly precipitation and soil water index for 50-yr recurrence interval over the Japanese archipelago, Sci. Online Lett. Atmos., 11, 118–123, https://doi.org/10.2151/sola.2015-028, 2015.

Saito, H., Nakayama, D., and Matsuyama, H.: Relationship between the initiation of a shallow landslide and rainfall intensity — duration thresholds in Japan, Geomorphology, 118, 167–175, https://doi.org/10.1016/j.geomorph.2009.12.016, 2010.

Salcedo-Sanz, S., Ghamisi, P., Piles, M., Werner, M., Cuadra, L., Moreno-Martínez, A., Izquierdo-Verdiguier, E., Muñoz-Marí, J., Mosavi, A., and Camps-Valls, G.: Machine learning information fusion in Earth observation: A comprehensive review of methods, applications and data sources, Inf. Fusion, 63, 256–272, https://doi.org/10.1016/j.inffus.2020.07.004, 2020.

Segoni, S., Battistini, A., Rossi, G., Rosi, A., Lagomarsino, D., Catani, F., Moretti, S., and Casagli, N.: Technical Note: An operational landslide early warning system at regional scale based on space-time-variable rainfall thresholds, Nat. Hazards Earth Syst. Sci., 15, 853–861, https://doi.org/10.5194/nhess-15-853-2015, 2015.

Shimada, U., Sawada, M., and Yamada, H.: Evaluation of the accuracy and utility of tropical cyclone intensity estimation using single ground-based Doppler radar observations, Mon. Weather Rev., 144, 1823–1840, https://doi.org/10.1175/MWR-D-15-0254.1, 2016.

Yin, G., Yoshikane, T., Yamamoto, K., Kubota, T., and Yoshimura, K.: A support vector machine-based method for improving real-time hourly precipitation forecast in Japan, J. Hydrol., 612, 128125, https://doi.org/10.1016/j.jhydrol.2022.128125, 2022.

Young, C. B., Nelson, B. R., Bradley, A. A., Smith, J. A., Peters-Lidard, C. D., Kruger, A., and Baeck, M. L.: An evaluation of NEXRAD precipitation estimates in complex terrain, J. Geophys. Res. Atmos., 104, 19691–19703, https://doi.org/10.1029/1999JD900123, 1999.

**Responses to Referee 3 (RC3)**

**RC3: Comment 1 and response**

*This paper analyzed > 7,500 landslides in a region of Japan and insisted that the landslide density would be high when the rainfall return period exceeded 100 years. This paper deals with an interesting topic; the interpretation of results is reasonable for me. I hope the authors consider the comments below to make this paper more attractive to readers.*

Thank you again for commenting on our manuscript. We sincerely appreciate your constructive suggestions that would improve our manuscript. Please see below how we will revise the original manuscript to consider your recommendations.

**RC3: Comment 2 and response**

*The authors assume the stable conditions of rainfall. The meaning of "100 years" would differ in changing climate conditions. I want the authors to consider and mention climate change. The first step may be to examine trends in rainfall.*

This is a very important observation. We agree and acknowledge that the 100-year rainfall return level may shift over time due to climate change. Therefore, in the revised manuscript, we will follow your recommendation and examine the possible alteration of the estimated 100-year rainfall return level due to climate change. We will first assess trends in the annual maxima series (AMS) of rainfall intensities for multiple durations we used for estimating the 100-year rainfall return level. To this end, we will employ non-parametric statistical tests for assessing the significance and magnitude of the possible trends in rainfall (e.g., the Mann-Kendall test and the Sen's slope estimator test). Then, we will carefully revise our manuscript to add the new trend analysis tests and highlight the possible alteration of the 100-year rainfall return level in the future due to climate change.

**RC3: Comment 3 and response**

*The authors analyzed using the return period of rainfall and did not mention the absolute amount (intensity) of rainfall. I am wondering whether the absolute amount of rainfall may be more important than the return period for understanding the distribution of the landslides.*

As explained in our manuscript (P2, L32–39 and P6, L 132–143), constraining the absolute amount (intensity) of rainfall responsible for all landslides (i.e., 7,676) triggered during the examined rainfall event is difficult due to the disparate hydromechanical responses of affected hillslopes to forcing rainfall. Therefore, in this study, we used multiple timespans from 1 to 72 h within a standardized period ($P_{std}$) of 3 days that accumulated the maximum rainfall amount during the triggering event to examine the relationship between rainfall information and landslide

density. In doing so, we intended to consider multiple combinations of rainfall durations that could represent the effective rainfall duration needed for triggering the various landslides.

If we consider the rainfall intensity maxima for a specific duration (e.g., 24, 48, or 72 h) recorded during the examined rainfall event as the meaning of absolute rainfall intensity, we could find a significant statistical correlation between landslide density and the absolute rainfall intensity (Table 1 and Fig. 3). This means that the absolute rainfall intensity could also be important for explaining the spatial distribution of landslide density. But, this correlation did not necessarily mean that landslide density increased with increased absolute rainfall intensity for a specific timespan (e.g., 24, 48, or 72 h). Indeed, as shown in Fig 4c, the landslide density metrics in two grid cells with similar slope conditions were different despite the similarity in the rainfall intensity for 24–72 h durations and slope conditions. This led us to conclude that all rainfall intensity maxima matter for landslide occurrence. Therefore, despite the absolute rainfall amount or intensity could explain the distribution of landslides from a statistical prospect, rainfall return level is a better proxy for landslide density as it can thoroughly assess the rainfall intensities for multiple timespans. We will clear this out better by improving the manuscript.

**RC3: Comment 4 and response**

*The results section includes not only "results" but also "discussion". It may be better to combine these two sections as the "results and discussion" section.*

Because combining the results and discussion sections may make the paper difficult to follow by readers, we believe that separated "results" and "discussion" sections may address our findings better. We will carefully revise the "results" section to avoid any possible preliminary discussion of the study results.

**RC3: Comment 5 and response**

*I guess there are several studies focusing on the same landslides because these landslides would affect a large-scale impact on this region. The authors did not mention the factor determining the density of the grids with any return periods of < 100 years. Are there any tips from the previous studies?*

We could find a few previous studies that focused on the same examined study case, but using different landslide inventories, such as Dou et al. (2020) and Ozturk et al. (2021). Both works used statistical machine-learning methods to investigate the importance of numerous predisposing factors in landslide occurrence. Their findings showed that rainfall is the main factor controlling landslide occurrence in our study area, followed by the slope and land use parameters. These findings provided useful insights about possible influence of terrain settings (i.e., slope and land cover) on landslide occurrence in the R/A grid cells with return periods < 100 years.

Therefore, in the revised manuscript, we will settle for improving the paragraph (P14 L329-

L333) to add the potential influence of terrain settings (e.g., land cover) on landslide occurrence when rainfall return levels are lower than 100 years.

**References**

Dou, J., Yunus, A. P., Bui, D. T., Merghadi, A., Sahana, M., Zhu, Z., Chen, C. W., Han, Z., and Pham, B. T.: Improved landslide assessment using support vector machine with bagging, boosting, and stacking ensemble machine learning framework in a mountainous watershed, Japan, Landslides, 17, 641–658, https://doi.org/10.1007/s10346-019-01286-5, 2020.

Ozturk, U., Saito, H., Matsushi, Y., Crisologo, I., and Schwanghart, W.: Can global rainfall estimates (satellite and reanalysis) aid landslide hindcasting?, Landslides, 18, 3119–3133, https://doi.org/10.1007/s10346-021-01689-3, 2021.

---

## Author Response (AR1)

We thank the Associate Editor for handling our manuscript. Also, we are grateful to the three referees for their insightful observations and critiques. Following their constructive comments, we carefully revised our manuscript to clarify the methods and significance of this research.

Hereafter, we provide detailed responses to all received comments. The comments of the

Associate Editor and the three Referees are in *italic black* font style. Our responses are in regular blue font style. The changes we made in the manuscript are in regular brown font style.

**8    Responses to the Associate Editor (EC1)**

**9    EC1: Comments and responses**

*Dear Authors,*

*We have now received three referee comments (RCs). Based on the RCs, major revisions may be*

*needed before the manuscript may be considered for publication.*

*Please respond to the three Referee Comments. RC2, in particular, provided detailed critiques and*

*suggestions for improving the manuscript.*

*Upload a revised manuscript and a detailed response to the RCs by March 10, 2023.*

*Best,*

*Sagy Cohen, Associate Editor*

Thank you again for handling our manuscript. We have considered the insightful comments of the three Referees to improve our manuscript.

We understand and respect the critiques given by RC2. However, we do feel that most of them originated from an intrinsic misunderstanding of our study hypothesis and methods, which might be due to an unclear explanation in the original manuscript. Therefore, we thoroughly improved our manuscript to avoid any possible future misunderstandings by readers.

Hereafter, we provide our responses to all observations of the referees to explain how we revised our manuscript to consider their constructive comments.

**Responses to Referee 1 (RC1)**

**RC1: Comment 1 and response**

*In the introduction part, the authors should clearly indicate the research gap and the novelty of*
*this research.*

Thank you again for assessing our manuscript. We have thoroughly revised the
Introduction section of our manuscript to clearly state the research gap, hypothesis, and novelty.
Revision: P2 L24–80

[revised manuscript text omitted]

**RC1: Comment 2 and response**

*The research object of this paper is mainly shallow landslides. It is recommended to highlight the*
*uniqueness of the research object in the abstract and introduction.*

We acknowledge that most landslides triggered during the examined rainfall event are
shallow (depth = 1 to 2 m), as indicated by Chigira et al. (2018). Still, some of the landslides could
be relatively deep, as we observed a few landslides with large areas (area > 10,000 m$^2$) in the FAD
of the landslide inventory (Figure 2 in Page 5).
Because we did not use any fundamental criteria to differentiate shallow landslides (e.g.,
area < 10,000 m$^2$ in Marc et al. (2019)) due to the unavailability of validation data (i.e., highresolution DEM data taken before and after the examined event), we believe that adding "shallow landslides" may cause some confusion for readers. Therefore, we prefer not to limit the study to shallow landslides.

**RC1: Comment 3 and response**

*In Figure 1b, the north arrow is missing.*

We added the missed north arrow in Figure 1b. Also, we added the missed label and unit in the color bar of Figure 1a.

Revision: Please see Figure 1 in P4 L105

**RC1: Comment 4 and response**

*In Figure 3, the contour of the study area should be added. The color bars in Figure 3 lack labels and units. Please check similar issues in other figures.*

We have added the contour lines of the study area in Figure 3 (please see Fig. RC1.1). However, we feel that the figure becomes unclear for readers as it overlays multiple different information (i.e., rainfall intensity, landslide distribution, TD, MLD, and contour lines). We believe that adding the contour lines may make the figure difficult to understand. Therefore, we prefer not to add it.

On the other hand, we added the missed labels and units in the color bars in Figure 3 and all other figures in the Supplement file.

Revision: Please see Figure 3 in P11

Revision: Please see Supplement file, P5–P10

**RC1: Comment 5 and response**

*The discussion part needs to be reorganized.*

We reorganized the discussion section in the revised manuscript with sub-sections to make it more accessible for readers. In section "4.1 Rainfall return levels govern landslide density", we discussed the key findings of our research. In section "4.2. Importance of considering rainfall return levels as explanatory for landslide spatial distribution", we showed why the conventional quantitative statistical relationships could not explicitly investigate rainfall controls on landslide density and the importance of comparing rainfall return levels for multiple timespans to understand landslide spatial distribution.

Revision: P16 L348–418

[revised manuscript text omitted]

**RC1: Comment 6 and response**

*Figure 4c: "TD 5.68 & MLD = 1.14" should be changed to "TD = 5.68 & MLD = 1.14".*
We re-created Figure 4 to correct this mistake.

**RC1: Comment 7 and response**

*Line 255: The 100-year rainfall anomaly was higher in the low landslide-density grid cell in P3 (Fig.*
*5i) than in the low landslide-density grid cell in P1 (Fig. 5c) (< 1.5 times). Why could the comparison*
*of the 100-year rainfall anomaly explain the substantial difference in landslide density between*
*the two grid cells (≈ 110 times for TD).*

It is worth noting that the 100-year rainfall anomaly was proposed in our study for setting
a quantitative reference that assesses the spatial disparity in rainfall return levels and their
relation to the variation in landslide density. Also, it reflects important information on the rarity
and extremity of rainfall intensity for multiple timespans, irrespective of the differences in rainfall
intensities. For instance, a 100-year rainfall anomaly for a 3-h timespan higher than 1 means that
the 3-h maximum rainfall intensity was extreme and rare compared to previously experienced 3-
h rainfall intensity in the study area as it has a return level of > 100-year return period. Thus, the
100-year rainfall anomaly can provide important information on the potential of the multiple
rainfall timespans to induce landslides, as high return level rainfall is generally needed for
landsliding (Iida, 1999; Segoni et al., 2015). Accordingly, it can be a standard method to compare
the potential of rainfall intensity maxima observed in the different R/A grid cells to trigger
landsliding, irrespective of the differences in rainfall intensity maxima.

We found that the 100-year rainfall anomaly was higher in the low landslide-density grid
cell in P3 (Fig. 5i) than in the low landslide-density grid cell in P1 (Fig. 5c). This means that rainfall
timespans in the former were more extreme (i.e., high potential to cause landslides) than those
experienced over the latter. Accordingly, the differences in the 100-year rainfall anomaly, which
dictate the potential of rainfall periods to cause landsliding, could explain the substantial
difference in landslide density over the two R/A grid cells.

Please note that this statement (i.e., "the comparison of the 100-year rainfall anomaly
could explain the substantial difference in landslide density between the two grid cells (≈ 110
times for TD)") was deleted from the revised manuscript to avoid any preliminary discussion of
our findings in the "Results" section, following the recommendation of RC3 (please see RC3:
Comment 4 and response).

[Figure]

**Figure RC1.1.** Spatial distribution maps of rainfall intensity maxima for 1 to 72 h timespans within the $P_{std}$ in mm/h, triggered landslides (grey polygons), and landslide density metrics (circles). The brown lines show the contour lines of the study area.

**References**

**References**

Chigira, M., Sixian, L., and Matsushi, Y.: Landslide disaster induced by the 2017 northern
Kyushu rainstorm, Disaster Prevention Research Institute Annals, 28-35 (in Japanese, with English
abstract) pp., 2018.

Iida, T.: A stochastic hydro-geomorphological model for shallow landsliding due to
rainstorm, Catena, 34, 293–313, https://doi.org/10.1016/S0341-8162(98)00093-9, 1999.

Marc, O., Gosset, M., Saito, H., Uchida, T., and Malet, J. P.: Spatial Patterns of Storm-
Induced Landslides and Their Relation to Rainfall Anomaly Maps, Geophys. Res. Lett., 46, 11167–
11177, https://doi.org/10.1029/2019GL083173, 2019.

Segoni, S., Battistini, A., Rossi, G., Rosi, A., Lagomarsino, D., Catani, F., Moretti, S., and
Casagli, N.: Technical Note: An operational landslide early warning system at regional scale based
on space-time-variable rainfall thresholds, Nat. Hazards Earth Syst. Sci., 15, 853–861,
https://doi.org/10.5194/nhess-15-853-2015, 2015.

**Responses to Referee 2 (RC2)**

**RC2: Comment 1 and response**

*The study relates a large data set of landslides with rainfall characteristics in Japan, using 7,500*
*landslides over an area of 400km². The study uses radar precipitation at 25km2 resolution with 1*
*to 72 h durations. Land cover and lithology are deemed homogenous in the study site.*
*A power-law distribution is used to identify the landslide size cutoff for moderate and large sizes.*
*Landslide densities are only calculated where slopes exceeded a threshold of 16.26 degrees (slopes*
*that include >90% of slides). Landslides are separated into total landslide density (TD), which*
*includes all the observations, and medium and large landslide size density (MLD), which includes*
*the slides greater that the size cutoff (>439 m2).*
*A standardized rainfall that accumulates maximum rainfall over 72h period is used as Pstd. Within*
*this Pstd, multiple time periods that record maximum intensities were also identified (1h to 72h).*
*That aided the authors to develop a rainfall intensity-duration relation threshold curves based on*
*I-D data.*
*Figure 3 presents a map of 1h to 72h maximum rainfall depths (25km2 resolution) along with TD*
*and MLDs. Higher landslide densities are observed where rainfall intensities are high.*
*More landslides occurred with rainfall exceeded 100 year return interval.*

Thank you for assessing our manuscript. We like to clarify a potential misunderstanding
about how we calculated landslide density in this study. Our study intended to examine whether
rainfall return levels govern landslide spatial distribution during rainfall events. Given that the
rainfall information was derived from a 5-km radar-driven gauge-adjusted precipitation dataset
(referred to as R/A), we calculated landslide density by considering the number of landslides that
occurred within each R/A grid cell. This is different from other studies that intended to examine
how landslide density varies with slope angle, and therefore they calculated landslide density by
counting the number of landslides that occurred within particular ranges of local hillslope angles
(e.g., Coe et al., 2004; De Rose, 2013; Prancevic et al., 2020).

So, differently from what is stated, "*Landslide densities are only calculated where slopes*
*exceeded a threshold of 16.26 degrees (slopes that include >90% of slides)*", landslide densities
considered the number of **all** landslides (for total landslide density "TD") and **all** landslides with
area > 439 m² (for medium and large landslides density "MLD") occurred within each R/A grid cell
(i.e., ≈ 25 km²). The threshold of 16.26° (considered in our study as a minimum slope threshold to
allow landsliding and referred to as $S_{threshold}$) was used to calculate the area of the R/A grid cells
where the slope > 16.26° (referred to hereafter as $A_{threshold}$). The two Landslide density metrics
were, therefore, calculated by dividing the number of landslides (i.e., **all** landslides for TD and **all**
landslides with an area > 439 m² for MLD) that occurred within each R/A grid cell by $A_{threshold}$
following the equation (1) and (2).

$$TD = \frac{\text{Total number of all landslides within the R/A grid cell}}{A_{threshold}} \qquad (1)$$

$$MLD = \frac{\text{Number of medium and large landslides within the R/A grid cell}}{A_{threshold}} \quad (2)$$

Such a normalization method is fundamental to reduce bias in the numbers of triggered landslides within the different R/A grid cells caused by the differences in the distribution of local topographic features (Prancevic et al., 2020), as landslides commonly occur in hilly and mountainous areas rather than plains (Lombardo et al., 2021). Therefore, it makes assessing the relationship between rainfall information and landslide densities in the R/A grid cells less biased by the differences in local topographic conditions. We note that such a normalization method has been also adopted in some previous works by considering a 10° slope as the minimum slope threshold for landsliding (Marc et al., 2019) or the slope at which > 90 % of landslides occurred (Prancevic et al., 2020).

In the revised manuscript, we rewrote section 2.3 to explain clearly the method of landslide density calculation. Additionally, we reorganized this session into two sub-sections for clarity reasons. Section 2.3.1 explains how we calculated the landslide density metrics. Section 2.3.2. describes the methods we followed in this research for investigating the relationships between the spatial pattern of landslide density and rainfall information.

Revision: P8 L190–211

**2.3.1. Landslide density**

The spatial distribution of triggered landslides over the study area can be described as a spatial variation of landslide density (i.e., number/km$^2$). Landslide density is generally calculated by counting the number of landslides that occurred within a specific area. Here, because we intended to reveal the potential control of rainfall return levels for multiple timespans derived from the R/A dataset on the variation of landslide density, we used the R/A grid cell ($\approx$ 25 km$^2$) as a sliding window to calculate landslide density. To count the number of landslides that occurred within each R/A grid cell, we converted the polygons data of landslide scars to points locating the centroid of each polygon. These numbers are generally biased by the non-uniformly distributed topographic features (i.e., hills, mountains, plains, lakes) within the different R/A grid cells because landslides commonly occur in hilly and mountainous areas rather than plains (Lombardo et al., 2021). To avoid such a possible bias, landslide density was calculated as the number of landslides within each R/A grid cell divided by the area of the R/A grid cell where the slope is higher than a threshold angle ($S_{threshold}$) assumed to be a minimum angle to allow landsliding. $S_{threshold}$ defines the threshold angle above which 90 % of landslides occurred (Prancevic et al., 2020) and was determined as 16.26° based on the DEM data analysis (Fig. S1).

Although medium and large landslides (landslides with area size exceeding the cutoff point of the FAD (439 m$^2$)) counted only 28.12 % of the total landslides, their areas represented more than 70 % of the total landsliding area (i.e., the total scar areas of the triggered landslides). Therefore, it is interesting to investigate rainfall controls on the density of total and only medium and large landslides. Accordingly, we computed two landslide density metrics, total landslide density (TD) and only medium and large landslide density (MLD), as the number of landslides per unit area (km$^2$), for each R/A grid cell using the following equations (1) and (2). Note these metrics represent averaged landslide density within the R/A grid cells.

$$TD = \frac{\text{Total number of all landslides within an R/A grid cell}}{A_{threshold}} \quad (1)$$

$$MLD = \frac{Number\ of\ medium\ and\ large\ landslides\ within\ an\ R/A\ grid\ cell}{A_{threshold}} \tag{2}$$

Where, $A_{threshold}$ is the area in km$^2$ of an R/A grid cell where the slope > S$_{threshold}$ (i.e., 16.26°).

**RC2: Comment 2 and response**

*Observations: P1, P2, P3-- can you clarify how the populations of landscape slopes similar in these*
*groups, do you report any statistics somewhere? Where are those populations? Are they identified*
*within each selected rainfall grid or can they be located in different rainfall grids?*

It is worth noting that each of the pairs (i.e., P1, P2, and P3) represents two R/A grid cells
with comparable local slope distributions within $A_{threshold}$ but different landslide density metrics
(i.e., TD and MLD). The selection of the three pairs was based on the distribution of local slope
conditions within $A_{threshold}$ of the different R/A grid cells rather than landslide data. In other
words, we examined all slope pixels (resolution = 10 m) in $A_{threshold}$ and did not limit the analysis
to only landslide slope pixels. By selecting these pairs, we intended to explicitly focus on rainfall
controls and avoid any possible influence of the non-uniformly distributed slopes within $A_{threshold}$
of the R/A grid cells on landslide occurrence.

The three pairs were selected by first comparing the distribution of slope conditions in
$A_{threshold}$ of all R/A grid cells (i.e., 23) using the Kruskal-Wallis static (Kruskal and Wallis, 1952) to
validate the existence of significant differences in local slope conditions. To better highlight these
differences, we provided a Figure showing the distribution of local slope degrees in $A_{threshold}$ of
the different R/A grid cells referred to in this figure by the corresponding TD (please see Fig. RC2.1).
Subsequently, we employed Dunn's post hoc test for detecting the R/A grid cells with a similar
mean rank sum of slopes, meaning similar slope conditions. We note that the result of Dunn's
test has been already shown in Table S1 in the Supplement file, as stated in our preprint (P8, L198).
From this result, we could find three pairs of R/A grid cells characterized by similar slope
conditions (as Dunn's test could not reject the null hypothesis) and different landslide density
metrics. Therefore, to explicitly reveal the controls of rainfall information on landslide density,
we mainly focused on these three pairs (i.e., P1, P2, and P3) as each pair of R/A grid cells includes
two R/A grid cells with comparable local slope distributions.

[Figure]

**Figure RC2.1.** Distribution of local slope degree within $A_{threshold}$ of the R/A grid cells. Note that the distributions are shown as box-and-whisker plots. The box delimitates the 25[th] and 75[th]

percentiles. The black line indicates the median. The red cross '+' displays the mean. The circles

'o' designate the outliers.

  In the revised manuscript, we rewrote section 2.3 to explain clearly how and why we selected the three pairs of R/A grid cells in this research.

Revision: P8 L212–231

**2.3.2. Relationships between the spatial pattern of landslide density and rainfall information**

Similar to previous studies (e.g., Chang et al., 2008), our investigation started by evaluating the statistical correlations between calculated landslide density metrics (TD and MLD) and rainfall intensity maxima for multiple timespans (1–72 h). We used Spearman's rank coefficient (ρ) to measure the non-parametric monotonicity of these relationships. In doing so, we intended to explore whether the developed statistical relationships can explicitly explain the rainfall controls on landslide density. Subsequently, we compared the variation in rainfall intensity maxima and their return levels and landslide density at the R/A grid cell scale.

Although the use of $A_{threshold}$ as a normalization method for calculating TD and MLD suppresses the influence of the non-uniformly distributed topographic features within the different R/A grid cells, still, these metrics can be biased by the non-uniformly distribution of local slopes within the $A_{threshold}$ as landslide occurrence also depends on hillslope steepness (Prancevic et al., 2020). Therefore, it is crucial to focus on R/A grid cells with comparable local slope distributions to explicitly investigate the potential control of rainfall intensity maxima and their return levels on landslide density. To this end, we first tested the differences in local slope angle distribution within $A_{threshold}$ of the different R/A grid cells using the

Kruskal-Wallis test (Kruskal and Wallis, 1952). Then, we employed Dunn's nonparametric pairwise test (Dunn, 1961) with a Bonferroni correction for the *p-value* for detecting the R/A grid cells with similar mean
rank sums of slopes within $A_{threshold}$ (similar slope conditions). Here, the null hypothesis assumes no
significant differences in the distribution of slope angles within the $A_{threshold}$ of the R/A grid cells.
Therefore, the *p-value* should be higher than a significant level of 5 % to accept the null hypothesis (Dinno,
2017). Accordingly, the pairwise R/A grid cells, where Dunn's test accepts the null hypothesis, would be
ideal examples for comparing the relation between rainfall intensity maxima and their return levels and
the variation of landslide density metrics.

Additionally, we rewrote a part of the Result section to present the results of Dunn's test
used for selecting the three pairs of R/A grid cells and integrated Figure RC2.1. in the revised
manuscript (Figure S3 in the Supplement Information) to provide the reader with clear
information on the non-uniformly distributed slopes within the different R/A grid cells.
Revision: P9 L247–254
The 23 R/A grid cells, where the triggered landslides were distributed, exhibited significant non-uniformly
distributed local slopes within $A_{threshold}$, as shown in Fig. S3, and confirmed by the rejection of the null
hypothesis of the Kruskal-Wallis test (*p-value* < 0.05). Applying Dunn's post hoc test, we could idealize
three pairs of R/A grid cells with comparable slope distributions within $A_{threshold}$, as Dunn's test could not
reject the null hypothesis (Table S1). These three pairs of R/A grid cells were referred to as P1, P2, and P3
and focused on hereafter to explicitly investigate the relation between rainfall intensity maxima and
landslide density (Fig. 4). Note we excepted three R/A grid cells where most landslides occurred in areas
affected by anthropogenic activities (e.g., slopes surrounding cropland and paddy field) from the Dunn's
post hoc test.

**RC2: Comment 3 and response**

*Lines 195-220: I'm not sure what the objective here, if one is interested to find out where rainfall*
*plays a stronger role, then shouldn't you go and investigate the local conditions (area, slope, soil*
*veg properties) of individual slides.*
Here, we compared the relation between rainfall intensity maxima and landslide density
in three pairs of R/A grid cells with comparable local slope distributions (i.e., P1, P2, and P3). We
intended to explore the potential relation between the rainfall intensity maxima and the spatial
variation of landslide density metrics (i.e., TD and MLD). In other words, we intended to
investigate whether landslide density necessary increased with the increase in rainfall intensity
maxima.
We agree that one of the methods is to investigate local conditions (e.g., slope, soil,
vegetation properties, etc.). However, there are mainly one or two controlling factors in some
specific regions which are worth exploring. In our study area in particular, two interesting
previous works have investigated the importance of multiple predisposing factors (e.g., slope,
land cover, elevation) in landslide occurrence using statistical machine-learning methods (Ozturk
et al., 2021; Dou et al., 2020). Both works showed that rainfall is the main factor controlling landslide occurrence in our study area, followed by the slope and land use parameters. These
findings were also consistent with the in-field observation of Chigira et al. (2018). It is worth
noting also that several previous works showed the feasibility to assess only rainfall conditions
for landslide prediction by exploring the spatial relation between rainfall conditions and landslide
density (Chen et al., 2013; Chang et al., 2008; Dai and Lee, 2001; Gao et al., 2017; Marc et al.,
2019), as rainfall is the main factor for landsliding. Given this, we mainly focused on rainfall
controls on landslide density in this study.
In the revised manuscript, we added the findings of Ozturk et al. (2021) and Dou et al.
(2020) to explain why we can focus on rainfall controls on landslide occurrence in the study area
while ignoring other predisposing factors.
Revision: P3 L86–94
If the landslides occurred in a homogeneous regolith, which reduces the likelihood of their link to complex
geotechnical site characteristics (Marc et al., 2019), the interpretation of the potential rainfall controls on
landslide occurrence would be possible. Indeed, most landslides triggered by the examined rainfall event
were shallow, affected mainly the soil mantle, and occurred on forested hillslopes with similar lithological
settings (granodiorite and pelitic schist) (Chigira et al., 2018). Accordingly, previous investigations of the
importance of multiple predisposing factors (e.g., rainfall, slope, elevation, land cover, etc.) in the
occurrence of these landslides using machine learning methods showed the outweighing of rainfall
conditions on the other predisposing factors (Dou et al., 2020; Ozturk et al., 2021). Thus, the examined
area provides an adequate test field to investigate the rainfall controls on landslide density because at
least the land cover and lithological settings of hillslopes can be deemed relatively homogenous.

*I think the selection process of P groups are based on some random selection routine, if you shuffle*
*these landslides into another set of 3 populations you may get all three look like P1 and P2 with*
*smaller differences in rainfall rate differences, then what would you do.?*
From this comment, we believe the Referee interpreted the selection of the three pairs as
it was based on a random selection from the landslide data. Very differently, the three selected
pairs of R/A grid cells were selected based on local slope distributions within the R/A grid cells.
Please see our response to your second comment (RC2: Comment 2 and response), where we
have cleared out how we selected the three pairs of R/A grid cells and explained the revisions we
made in the revised manuscript to avoid any potential future misunderstandings.
*I also could not figure out what those two different groups are within each plot in Figure 4. Why*
*do the gray symbols have smaller landslide densities than red symbols? I think those were referred*
*to as "pairs" but not sure how paired and why with different densities? Beyond all what is the*
*purpose of pairing.*
In Figure 4, each plot compared rainfall intensity for multiple timespans (i.e., rainfall
intensity maxima) recorded in two R/A grid cells with comparable slope distributions (for the

$A_{threshold}$), but different numbers of landslides as can be revealed by the two landslide density
metrics (i.e., TD and MLD). For instance, in Fig. 4a, the gray symbols reflect the rainfall intensity
maxima recorded in the R/A grid cell where TD = 0.05 and MLD = 0 landslides/km². The red dots
reflect the rainfall intensity maxima observed in the R/A grid cell where TD = 35.61 and MLD =
11.98 landslides/km². The black line showed the average rainfall intensity maxima in the two R/A
grid cells in comparison.

The pairing approach we used in this paper aimed at selecting the R/A grid cells with
comparable slope conditions to avoid any possible influence of the differences in slope conditions
on landslide density and explicitly focus on rainfall controls, as we explained in our response to
your second comment (RC2: Comment 2 and response).

To avoid any potential future misunderstandings by readers, we changed the title and
legend of Figure 4 to show clearly that the red and gray points are rainfall intensity maxima from
R/A grid cells with different landslide density metrics.
Revision: Please see Figure 4 in P12 L280

**RC2: Comment 4 and response**

*Rainfall data is very coarse for a rugged terrain to obtain any detailed and new science with*
*respect to landslide process understanding and how rainfall controls it. The study may be useful*
*for regional early warning systems, though still very coarse.*

We agree that high-resolution rainfall data would provide more detailed information on
spatial rainfall patterns. However, long-term gridded rainfall data with a spatial resolution finer
than 5 km, needed in our study to estimate rainfall return levels, is currently unavailable in Japan.
Indeed, the R/A dataset used in this study is, so far, the highest-resolution and most reliable long-
term gridded precipitation data available. Due to its relatively high resolution, long-term records,
and accuracy, several studies used the R/A dataset as referential data for analyzing localized
heavy rainfall (e.g., Kato, 2020; Hirockawa et al., 2020; Saito and Matsuyama, 2015), evaluating
precipitation forecasts and estimates (e.g., Kubota et al., 2009; Iida et al., 2006; Yin et al., 2022),
and constraining empirical relationships between rainfall information and landslide occurrence
(e.g., Saito et al., 2010; Marc et al., 2019; Ozturk et al., 2021). All these works showed the
usefulness of the R/A precipitation product in capturing the spatial pattern of extreme rainfall
events experienced over the Japanese archipelago, as it could sufficiently resolve mesoscale
convective systems (Hirockawa et al., 2020).
Interestingly, Ozturk et al. (2021) evaluated the performance of a coarsened R/A dataset
to ≈ 10-km resolution in landslide forecasting using a logistic regression model and showed a
comparable performance between the 5-km and 10-km R/A dataset, meaning that the spatial
rainfall pattern over the mountainous study areas Ozturk et al. (2021) focused on can be
satisfactorily captured even with a 10-km spatial resolution R/A data. Therefore, as our objective was to explore the spatial relation between rainfall characteristics and landslide density, rather than explicitly examine the landsliding process of each of the triggered landslides, we believe that a resolution of 5 km could be sufficient due to its performance in capturing the spatial pattern of the studied rainfall event and given the unavailability of alternative product with finer resolution and long-term records.

*How do you take the next step from coarse-grain analysis to finer scale hazard mapping?*

We believe that the R/A data can be downscaled to finer resolution by employing machine learning and data fusion methods (e.g., Peleg et al., 2018; Salcedo-Sanz et al., 2020) to address finer scale hazard analysis. However, several drawbacks can limit the application of these methods, such as the need for dense rain gauges network over mountainous regions, which is generally difficult to obtain. We believe that rainfall data downscaling is another research issue that needs to be addressed in detail in the future and is beyond the objective of the current study.

**RC2: Comment 5 and response**

*What is the point of Figure 5, what is the question you are trying to address?*

Thank you for this important question that leads us to notice an insufficient explanation about investigating rainfall return levels in our preprint (in particular, Figure 5). The question we tried to address in Figure 5 is to investigate whether rainfall return levels constrain landslide density during the examined rainfall event. In other words, we tried to evaluate whether landslide density increased with the increase in rainfall return levels. The use of the return levels in this study was motivated by the fact they can indirectly evaluate whether rainfall is likely to trigger landslides without the need for historical landslide records in the targeted regions, as shown in multiple previous works (e.g., Tsunetaka 2021).

We revised the Introduction section to clarify the motivation beyond investigating the relation between rainfall return levels and landslide density (Figure 5).

Revision: P2 L43–66

So far, we still lack information on the best rainfall variable(s) constraining the landslide spatial pattern during rainfall events. Some works showed increased landslide density with the increase in total rainfall amount, rainfall duration, the maximum rainfall amount for short durations (e.g., 3, 12, 24 h), or antecedent rainfall (Marc et al., 2018; Chen et al., 2013; Chang et al., 2008; Dai and Lee, 2001; Abanco et al., 2021). Other studies demonstrated that normalized rainfall amounts for specific timespans (e.g., 2, 24, 48 h) by the mean annual precipitation (Ko and Lo, 2016) or the 10-year return period rainfall amount (Marc et al., 2019), which explain the landscape coevolution with local climate (Benda and Dunne, 1997; Iida, 1999), are better predictors for landsliding.

On the other hand, these statistical relationships allow the development of rainfall-based empirical models for predicting the number of landslides likely to be triggered by future rainfall events (e.g., Chang et al.,

2008). However, their development and extrapolation to other regions are challenging. Constraining any
spatial relationship requires comprehensive landslide inventories that contain sufficient landslides for an
adequate statistical analysis. However, this need is extremely difficult to fulfill (Marc et al., 2018; Emberson
et al., 2022). Furthermore, the constrained quantitative relationships are very sensitive to the landslide
records and the characteristics of respective triggering rainfall events used in the statistical analysis.
Therefore, they are case-specific and cannot always be extrapolated to predict the number of landslides
likely to be triggered by future rainfall events, even in the same region (e.g., Gao et al., 2018).

For a given rainfall event, the return period of any rainfall episode with specific duration and intensity can
be assessed using the Intensity-Duration-Frequency (IDF) curves, which are equipotential lines of
probabilities linking rainfall durations and maximum intensities from long-term records (Chow et al., 1988).
This information can potentially evaluate whether a rainfall event is likely to cause landslides as a high
rainfall return level (i.e., rare rainfall event) is generally considered a proxy for the critical rainfall
conditions triggering landslides (Frattini et al., 2009; Griffiths et al., 2009; Segoni et al., 2015, 2014; Iida,
2004). Several studies showed the usefulness of considering rainfall return levels to indirectly evaluate the
potential of a forecast rainfall to trigger landslides without the need for historical landslide records in the
targeted region (e.g., Kim et al., 2021; Tsunetaka, 2021; Vaz et al., 2018). Still, the potential relation
between the spatial patterns of rainfall return levels and landsliding remains unrevealed.

Also, we revised the Results section to clarify the point and outcomes of Figure 5 better.
Revision: P13 L282–335

[revised manuscript text omitted]

*As far as I understood you have some randomly selected data pairs with different landslide*
*densities and they seem to show some narrow range of variable ID trends, but this is expected*
*isn't it.*
Sorry, you misunderstood how we selected the three pairs of R/A grid cells. The selection
of these pairs was based on local slope distributions within the R/A grid cells rather than a random
selection of landslide data. Please see our response to your second and third comments for more
explanation (RC2: Comment 2 and response, RC2: Comment 3 and response).
*Another point I did not understand—in Figs 3 and 4, do each of the circles average many points*
*with different landslide densities?*
Fig. 3 shows the spatial distribution of rainfall intensities for multiple timespans, triggered
landslides, and landslide density metrics. White circles designate the TD in corresponding R/A grid
cells. Black circles indicate the MLD in corresponding R/A grid cells.
No, in Fig. 4, each plot compared rainfall intensities for multiple timespans recoded in two
R/A grid cells with comparable slope distributions (for the $A_{threshold}$), but different numbers of
landslides as can be revealed by the two landslide density metrics (i.e., TD and MLD). So, the
circles (red and gray) are the rainfall intensities for multiple timespans recorded in two R/A grid
cells. For instance, in Fig. 4a, the gray symbols reflect the rainfall intensities for multiple timespans
recorded in the R/A grid cell where TD = 0.05 and MLD = 0 landslides/km$^2$. The red dots reflect
the rainfall intensities for multiple timespans recorded in the R/A grid cell where TD = 35.61 and
MLD = 11.98 landslides/km$^2$. The black line showed the average of rainfall intensities between
the two R/A grid cells in comparison.
To avoid any potential future misunderstandings by readers, we changed the title and
legend of Figure 4 to show clearly that the red and gray points are rainfall intensity maxima from
R/A grid cells with different landslide density metrics.
Revision: Please see Figure 4 in P12 L280
**RC2: Comment 6 and response**

*Not having a clear research question and/or hypotheses makes it difficult to follow this paper.*
Our scientific question was to investigate the potential relation between rainfall return
levels for multiple timespans, which characterize the temporal rainfall pattern, and the spatial
pattern of landslide distribution during the examined triggering rainfall event (i.e., landslide
density spatial pattern). In other words, we intended to assess whether the spatial variation of
landslide density during the examined triggering rainfall event is governed by rainfall return levels.
We understand your concern about the clarity of the research question and hypothesis.
Therefore, following this comment and the comment of RC1, we have thoroughly revised the
Introduction section to improve the research hypothesis and question statement. Please see our response to RC1's comment (RC1: Comment 1 and response), where we explained how we
improved the introduction section.

*In addition, the methods rely on some comparisons of three similar slope populations (P1,2,3), and*
*pairing of data among them, the purpose of which was not clear.*
 Sorry, you misunderstood how and why we select the three pairs of R/A grid cells with
similar slope conditions. Please see our response to your second and third comments for more
explanation (RC2: Comment 2 and response, RC2: Comment 3 and response).

*If the whole point of the paper is to show that rainfall patterns and return intervals matter, that*
*is no surprise to anyone, that is why those intensity-duration thresholds were used for nearly a*
*century.*
 First, it is worth noting the existence of two empirical approaches for quantifying rainfall
characteristics that triggered landslides. The first approach is the traditional intensity-duration
(ID) thresholds that determined the minimum rainfall conditions necessary for likely triggering
landslides. The second approach, mainly used in this paper, relates the spatial variation of
landslide density with rainfall information beyond the ID thresholds.
 The objective of this paper was to primarily investigate whether the spatial patterns of
rainfall return levels govern the variation of landslide density during rainfall events. We showed
that landslide density is constrained by the return levels of rainfall for multiple timespans rather
than rainfall intensities. Our finding is different from other studies' findings that related the
spatial variation of landslide density to the variation of a single rainfall variable for a specific
timespan. Also, this is different from the ID thresholds that generally linked the occurrence of
landslides to specific rainfall conditions in terms of intensity and duration. So, given this, we
believe that the findings of our paper are novel and addressed a significant gap in the
understanding of rainfall controls on landslide density.

*In addition, the rainfall data is at 5km spatial resolution, which for mountain ranges, is very coarse,*
*and radar rainfall is usually not a good option for estimating mountain rainfall.*
 We are aware of the intrinsic drawbacks of weather radars in reliably observing
precipitation, which could be attributed to various meteorological, topographic, and technical
factors (e.g., beam blockage, ground clutter, anomalous beam propagation, and range effects)
(e.g., Borga et al., 2022). Therefore, we agree with the Referee's statement: "*radar rainfall is*
*usually not a good option for estimating mountain rainfall.*" However, we believe this is the case
for the raw uncorrected radar-driven precipitation data (e.g., Young et al., 1999). Differently, the
R/A dataset used in this study was processed by a quality control algorithm involving various
correction procedures for precipitation observation errors (Makihara, 2000; Hotta, 2018; Nagata,
2011). For instance, ground clutter and beam blockage due to mountains are corrected using a 2- km Pseudo Constant Altitude Plan Position Indicator (PCAPPI) that processes echo intensity data from multiple elevation angles. Additionally, the R/A product involves a Gauge-adjustment algorithm that calibrates precipitation estimates with gauge measurements. These correction procedures made the R/A product valuable for providing reliable rainfall estimates over the mountainous areas in Japan, which cannot be captured by rain gauged due to a sparse network. Therefore, it is often used as benchmark rainfall data in multiple studies over mountainous areas (please see RC2: Comment 4 and response).

It is worth noting, finally, that several previous studies showed the usefulness of corrected radar-driven precipitation datasets in observing the rainfall over mountains (e.g., Germann et al., 2006; Shimada et al., 2016; Nelson et al., 2016; Marra et al., 2022). Therefore, we believe that the R/A product used in our study provides reliable rainfall estimates over the mountainous areas in Japan.

In the revised manuscript, we added further information on the processing algorithm of the R/A dataset used for correcting rainfall observation errors. Also, we have added some references that proved the usefulness of the R/A product in multiple hydrological studies.

Revision: P6 L132–147

We employed the radar/rain gauge analyzed (R/A) precipitation dataset to examine the spatiotemporal pattern of the triggering rainfall and derive the return levels of rainfall intensities for multiple timespans in the Intensity Duration Frequency (IDF) curves. The R/A dataset is a gridded hourly precipitation product developed by the Japan Meteorological Agency (JMA) based on 5-minutely reflected echo intensities and doppler velocities of 46 C-band radars (Nagata, 2011). The processing algorithm of this product includes three steps. First, accumulated radar echo intensity data were processed by a quality control algorithm for correcting precipitation observation errors attributed to various meteorological, topographic, and technical factors (e.g., beam blockage, ground clutter, anomalous beam propagation, and range effects) (Makihara, 2000). Subsequently, the hourly accumulated corrected radar data were adjusted to rainfall measurements obtained from local rain gauges to produce accurate Quantitative Precipitation Estimates (QPE). Finally, the calibrated QPE from the 46 radars were processed and assembled to derive nationwide hourly precipitation maps that compose the R/A product (Makihara, 2000; Nagata, 2011). This correction and processing scheme made the R/A dataset the most reliable long-term precipitation data over the Japanese archipelago. Accordingly, it has often been used as referential data for analyzing localized heavy rainfall (e.g., Kato, 2020; Hirockawa et al., 2020; Saito and Matsuyama, 2015), evaluating precipitation forecasts and estimates (e.g., Kubota et al., 2009; Iida et al., 2006; Yin et al., 2022), and constraining empirical relationships between rainfall information and landslide occurrence (e.g., Saito et al., 2010; Marc et al., 2019; Ozturk et al., 2021).

Also, we have added a paragraph to explain why the use of the R/A product in this study
is unavoidable.

Revision: P6 L155–158

Although the downscaling stage degrades the spatial details of rainfall events, it is unavoidable in this study
due to the requirement of long-term rainfall data in investigating rainfall return levels. Still, the
downscaled R/A dataset (i.e., 5-km resolution) can capture spatial rainfall patterns over the examined
region as it could sufficiently resolve mesoscale convective systems that resulted in most heavy rainfall
events in Japan (Hirockawa et al., 2020).

*And finally, which is probably more important than any of the comments I made above, besides*
*local slopes, the authors have not factored in elevation in their analysis. Elevation is also a good*
*predictor of rainfall and variations in soils and vegetation. They used a slope threshold in their*
*analysis to select landslides but a quick grouping by elevation would probably reveal a strong*
*elevation control.*

It is worth recalling that the slope threshold (16.26°) was used only for deriving normalized
landslide densities over the R/A grid cells while accounting for the number of **all** landslides (for
TD) and **all** landslides with area > 439 $m^2$ (for MLD).

Of course, we agree that the elevation can have a strong control on landslide occurrence
in addition to other predisposing factors for landslide occurrence (e.g., slope, land cover, rainfall,
etc.). However, there are mainly one or two controlling factors in some specific regions which are
worth exploring. For our study case in particular, Ozturk et al. (2021) evaluated the importance
of multiple predisposing factors for landslide occurrence, including elevation and rainfall, using
multivariate logistic regression. Their findings indicated that the rainfall information is the main
control for the spatial distribution of triggered landslides, followed by the slope parameter. On
the other hand, the elevation parameter was found to be very less important in controlling
landslide occurrence according to their findings.

To further assess how landslide occurrence varies with elevation, we have plotted the
histograms of landslide elevations (i.e., 7,676 landslides) from a 10-m DEM (please see Figure
RC2.2.). We found that the landslides occurred in hillslopes with a wide range of elevation from ≈
50 to ≈ 800 m a.s.l. Although most of the landslides occurred in hillslopes with an elevation in the
range of ≈ 50 to ≈ 600 m a.s.l., still, this elevation range is wide, meaning that landslide do not
preferentially occurred on hillslopes with a specific elevation.

Given this, we believe that the elevation has a weak control on the spatial distribution of
the landslides we focused on in this study.

[Figure]

**Figure RC2.2.** Non-cumulative (gray histogram) and cumulative (black line) frequency distribution
of landslide elevations (bins = 500). Note that landslide elevations were calculated as the median
of DEM pixel values at landslide scars.

In the revised manuscript, we added the findings of Ozturk et al. (2021) and Dou et al.
(2020) to explain why we can focus on rainfall controls on landslide occurrence in the study area
while ignoring other predisposing factors.
Revision: P3 L86–94
If the landslides occurred in a homogeneous regolith, which reduces the likelihood of their link to complex
geotechnical site characteristics (Marc et al., 2019), the interpretation of the potential rainfall controls on
landslide occurrence would be possible. Indeed, most landslides triggered by the examined rainfall event
were shallow, affected mainly the soil mantle, and occurred on forested hillslopes with similar lithological
settings (granodiorite and pelitic schist) (Chigira et al., 2018). Accordingly, previous investigations of the
importance of multiple predisposing factors (e.g., rainfall, slope, elevation, land cover, etc.) in the
occurrence of these landslides using machine learning methods showed the outweighing of rainfall
conditions on the other predisposing factors (Dou et al., 2020; Ozturk et al., 2021). Thus, the examined
area provides an adequate test field to investigate the rainfall controls on landslide density because at
least the land cover and lithological settings of hillslopes can be deemed relatively homogenous.

*All in all, the paper left me with no new information. If the authors would want to salvage this*
*paper, they would probably reconsider a set of new methods and pose clear questions and*
*objectives.*
We respect your critiques. However, we feel that most of them originated from an intrinsic
misunderstanding of the research methods, especially the method of landslide density calculation
and pairs selection. Considering the research objective was to mainly investigate whether rainfall
return levels govern landslide spatial distribution (i.e., or density), we believe that the methods
used in our study could sufficiently address the research question.

Finally, we apologize for any misunderstandings that might be originated from unclear
explanations of the research methods and hypothesis in the original manuscript. We substantially
revised the manuscript to state our research question and hypothesis better and improve the
presentation of the methods used in this study. We hope the current revised manuscript
addressed our research objective and findings clearly.


**Responses to Referee 3 (RC3)**

**RC3: Comment 1 and response**

*This paper analyzed > 7,500 landslides in a region of Japan and insisted that the landslide density would be high when the rainfall return period exceeded 100 years. This paper deals with an interesting topic; the interpretation of results is reasonable for me. I hope the authors consider the comments below to make this paper more attractive to readers.*

Thank you again for commenting on our manuscript. We sincerely appreciate your constructive suggestions that improved our manuscript. Please see below how we revised the original manuscript to consider your recommendations.

**RC3: Comment 2 and response**

*The authors assume the stable conditions of rainfall. The meaning of "100 years" would differ in changing climate conditions. I want the authors to consider and mention climate change. The first step may be to examine trends in rainfall.*

Thank you for this very important observation. It is indeed interesting to see whether the 100-year rainfall return level is already subject to climate change effect. Therefore, in the revised manuscript, we followed your recommendation and examined the possible alteration of the estimated 100-year rainfall return level due to climate change. We first assessed trends in the annual maxima series (AMS) of rainfall intensities for multiple durations we used for estimating the 100-year rainfall return level. To this end, we employed two non-parametric statistical tests for assessing the significance and magnitude of the possible trends in rainfall (i.e., the Mann-Kendall test and the Sen's slope estimator test). Then, we carefully added the outcomes of these two tests in the "Results" section.

The methods of the trend analysis were integrated in the Material and Methods section of the revised manuscript.

Revision: P7 L181–188

Although the Gumbel distributions may well fit the observed rainfall AMS based on the KS test, this does not mean that the derived IDF curves do not shift over time (i.e., stationary) due to climate change (Slater et al., 2021). It is, therefore, crucial to test the stationarity assumption in the Gumbel model parameters by assessing the existence of trends in rainfall AMS during the examined period. To this end, we employed the Mann-Kendall and Sen's slope tests, two non-parametric statics frequently applied in hydro-meteorology for trend analysis (e.g., Yan et al., 2018). The Mann-Kendall test assesses the significance of trends in rainfall (Mann, 1945; Kendall, 1975), while Sen's slope test quantifies the magnitude of these trends if exist (Sen, 1968). The null hypothesis of the Mann-Kendall test assumes no trends. Therefore, a *p-value* less than a significance level of 5 % would imply the existence of a significant trend in rainfall AMS.

We have also provided two new figures in the Supplement file showing the results of the
Mann-Kendall and Sen's slope tests.
Revision: Please see Supplement file, P11- P12
We note that these two tests showed a spatial heterogeneity of the significance and
magnitude of trends in rainfall annual maxima series for multiple timespans that need a detailed
investigation of its drivers. Given that the main objective of this paper is to investigate the relation
between rainfall return levels and landslide density, we avoided detailed analysis of the trend
tests as it is beyond the objective of the current study. Accordingly, the outcomes of the trend
analysis were briefly integrated in the Results section of the revised manuscript as shown below.
Revision: P13 L287–295
The Mann-Kendall and Sen's slope tests showed a spatial heterogeneity in the significance and magnitude
of trends in observed rainfall AMS (Figs. S10 and 11). Specifically, some R/A grid cells in the western part
of the study area showed statistically significant positive rainfall trends at the 95 % significance level, as
the Mann-Kendall rejected the null hypothesis ($p$-value < 0.05). Other R/A grid cells exhibited no significant
trends, especially for short-duration rainfall intensities (Fig. S10a–c), where Mann-Kendall accepted the
null hypothesis ($p$-value > 0.05). The increasing trends could be attributed to the climate change effect and
indicated that the rainfall IDF curves developed for the examined region are already subject to climate
change and may be altered in the future due to the persistent effect of climate change. Still, they could
provide valuable information about the return levels of the rainfall intensity maxima characterizing the
examined rainfall event.

**RC3: Comment 3 and response**

*The authors analyzed using the return period of rainfall and did not mention the absolute amount*
*(intensity) of rainfall. I am wondering whether the absolute amount of rainfall may be more*
*important than the return period for understanding the distribution of the landslides.*
Thank you for this important question. As explained in our revised manuscript (P3, L67–
72 and P7, L160–170), determining the absolute amount (intensity) of rainfall responsible for all
landslides (i.e., 7,676) triggered during the examined rainfall event is difficult due to the disparate
hydromechanical responses of affected hillslopes to forcing rainfall. Therefore, in this study, we
used multiple timespans from 1 to 72 h within a standardized period ($P_{std}$) of 3 days that
accumulated the maximum rainfall amount during the triggering event to examine the
relationship between rainfall information and landslide density. In doing so, we intended to
consider multiple combinations of rainfall durations that could represent the effective rainfall
duration needed for triggering the various landslides.
If we consider the rainfall intensity maxima for a specific duration (e.g., 24, 48, or 72 h)
recorded during the examined rainfall event as the meaning of absolute rainfall intensity, we
could find a significant statistical correlation between landslide density and the absolute rainfall
intensity (Table 1 and Fig. 3). This means that the absolute rainfall intensity could also be important for explaining the spatial distribution of landslide density. But, this correlation did not
necessarily mean that landslide density increased with increased absolute rainfall intensity, as we
observed grid cells with similar rainfall intensities but different landslide density. The landslide
density differed even for grid cells with comparable local slope distributions and rainfall
intensities, as shown in as shown in Fig 4c. This led us to conclude that rainfall intensity (i.e.,
absolute rainfall) do not necessarily constrain landslide density. On the other hand, landslide
density over the examined grid cells increased by the increase in rainfall return levels (Fig 5c, f).
Therefore, the results of our investigation showed that the landslide density is constrained by
rainfall return levels, rather than rainfall intensities.

We have thoroughly revised the Results section to clarify why we concluded that landslide
density is constrained by rainfall return levels rather than rainfall intensities.

Revision: P10 L262–273

Importantly, even with comparable rainfall intensities and slope distributions, landslide density over two
R/A grid cells could be different (Fig. 4c). Unlike the observations in P1 and P2, rainfall maxima recorded
for 12–72 h over the two R/A grid cells in P3 (Fig. 4c) were similar. The R/A grid cell with higher landslide
density experienced little higher rainfall intensity maxima for 1–6 h timespans than those recorded in the
R/A grid cell with lower landslide density. But, the differences in these rainfall intensity maxima were slight
(≈ 1.15 times) compared to those observed between the paired R/A grid cells in P1 and P2. Because P1 and
P2 paired two of the R/A grid cells with the lowest landslide density metrics during the examined rainfall
event with two of the R/A grid cells with the highest landslide density metrics, the differences in landslide
density metrics were much more pronounced than that observed over the R/A grid cells in P3 (≈ 3.5 times
for TD). However, the R/A grid cell with higher landslide density in P3 indicated the fifth highest TD (20.91
landslides/km$^2$) and MLD (5.65 landslides/km$^2$) in the total of 23 R/A grid cells (Fig. S3), being a sufficiently
high landslide density. Given this, the results in P3 indicated that differences in rainfall intensities and slope
distributions (i.e., topography) do not necessarily constrain landslide density.

Revision: P13 L305–313

Interestingly, despite the comparable rainfall intensities and slope distributions within the R/A grid cells in
P3 (Fig. 4c), return levels of short-duration rainfall intensity maxima differed, as for the landslide density
metrics (Fig. 5c and f). The return levels of rainfall intensity maxima in both R/A grid cells exceeded the
100-year return periods only for some timespans and shared comparable return levels for the rainfall
intensity maxima at 12–72 h. Still, the rainfall return levels for 1–6 h-intensities in the high landslide density
R/A grid cell (Fig. 5f) were higher than those observed in the R/A grid cells with lower landslide density
(Fig. 5c). For instance, the return level of 3-h rainfall intensity exceeded the 100-year return period in the
R/A grid cell with TD = 20.91 landslides/km$^2$ (Fig. 5f), but it was in the order of 50-year return period in the
R/A grid cell with TD = 5.68 landslides/km$^2$ (Fig. 5c). Therefore, the results in P3 showed that the landslide
density metrics over an R/A grid cell increased with the increase in rainfall return levels, rather than rainfall
intensities.

 **RC3: Comment 4 and response**

*The results section includes not only "results" but also "discussion". It may be better to combine*
*these two sections as the "results and discussion" section.*

Because combining the results and discussion sections may make the paper difficult to
follow by readers, we believe that separated "results" and "discussion" sections may address our
findings better.

We carefully revised the "results" section to avoid any possible preliminary discussion of
the study results. We removed some sentences (e.g., "This means that the disparities in rainfall
return levels could be the cause for the relative difference in landslide density between the two
paired grid cells.", "the comparison of the 100-year rainfall anomaly could explain the substantial
difference in landslide density between the two grid cells (≈ 110 times for TD)") that interpreted
our results were removed from the "results" section. We believe that now the Results section
only presents the findings of the current study.
Revision: P9 L233–343

[revised manuscript text omitted]

**RC3: Comment 5 and response**

*I guess there are several studies focusing on the same landslides because these landslides would*
*affect a large-scale impact on this region. The authors did not mention the factor determining the*
*density of the grids with any return periods of < 100 years. Are there any tips from the previous*
*studies?*
We could find a few previous studies that focused on the same examined study case, but
using different landslide inventories, such as Dou et al. (2020) and Ozturk et al. (2021). Both works
used statistical machine-learning methods to investigate the importance of numerous
predisposing factors in landslide occurrence. Their findings showed that rainfall is the main factor
controlling landslide occurrence in our study area, followed by the slope and land use parameters.
These findings provided useful insights about possible influence of terrain settings (i.e., slope and
land cover) on landslide occurrence in the R/A grid cells with return periods < 100 years.
Therefore, in the revised manuscript, we integrated the findings of these two important
works to add the potential influence of terrain settings (e.g., land cover) on landslide occurrence
when rainfall return levels are lower than 100 years.
Revision: P17 L385–394
Last, it is worth noting that landslides occurred even when rainfall did not reach the 100-year return level
at any of the examined timespans (Fig S12 b, e, f). However, landslide density over these grid cells (i.e.,
grid cells where rainfall did not reach the 100-year return level) was considerably low (≈ 0.4–1.5
landslides/km$^2$ in terms of TD) compared with most other grid cells. Dou et al. (2020) and Ozturk et al.
(2021) used statistical machine-learning methods to investigate the importance of numerous predisposing
factors in landslide occurrence by the examined rainfall event. Their findings showed that rainfall is the
main factor controlling landslide occurrence in our study area, followed by the slope and land use
parameters. Accordingly, landslide occurrence over these grid cells during the examined rainfall event
could be constrained by terrain settings (e.g., land cover) as the rainfall return levels were low. Therefore,
landslides can occur even if rainfall return levels do not reach the 100-year return period but with
substantially low density. In any case, comparing rainfall return levels in the IDF curves can explain the
substantial differences in landslide density due to considering multiple return periods.